# Cyclin B1 scaffolds MAD1 at the kinetochore corona to activate the mitotic checkpoint

Lindsey A Allan[1], Magda Camacho Reis[1], Giuseppe Ciossani[2], Pim J Huis in 't Veld[2] (iD),
Sabine Wohlgemuth[2], Geert JPL Kops[3] (iD), Andrea Musacchio[2] (iD) & Adrian T Saurin[1,*] (iD)

## Abstract

**Cyclin B:CDK1 is the master kinase regulator of mitosis. We show here that, in addition to its kinase functions, mammalian Cyclin B also scaffolds a localised signalling pathway to help preserve genome stability. Cyclin B1 localises to an expanded region of the outer kinetochore, known as the corona, where it scaffolds the spindle assembly checkpoint (SAC) machinery by binding directly to MAD1.** *In vitro* **reconstitutions map the key binding interface to a few acidic residues in the N-terminal region of MAD1, and point mutations in this sequence abolish MAD1 corona localisation and weaken the SAC. Therefore, Cyclin B1 is the long-sought-after scaffold that links MAD1 to the corona, and this specific pool of MAD1 is needed to generate a robust SAC response. Robustness arises because Cyclin B1:MAD1 localisation loses dependence on MPS1 kinase after the corona has been established, ensuring that corona-localised MAD1 can still be phosphorylated when MPS1 activity is low. Therefore, this study explains how corona-MAD1 generates a robust SAC signal, and it reveals a scaffolding role for the key mitotic kinase, Cyclin B1:CDK1, which ultimately helps to inhibit its own degradation.**

**Keywords** Cyclin B1; kinetochore corona; MAD; RZZ complex; spindle assembly checkpoint

**Subject Category** Cell Cycle

The EMBO Journal (2020) 39: e103180

See also: **C Conde & R Gassmann** (June 2020)

## Introduction

During mitosis, all duplicated chromosomes must attach correctly to microtubules so they can segregate properly when the cell divides. This attachment is mediated via the kinetochore, which is a giant molecular complex assembled on chromosomes at the centromere (Musacchio & Desai, 2017). As well as attaching to microtubules, the kinetochore must also regulate this process to ensure it occurs correctly. One aspect of this regulation involves the activation of the mitotic checkpoint, otherwise known as the spindle assembly checkpoint (SAC), which blocks mitotic exit until all kinetochores have attached to microtubules. The principle of the SAC is that each unattached kinetochore acts as a factory to produce an inhibitor of mitotic exit, known as the mitotic checkpoint complex or MCC (for further molecular details, see Corbett, 2017). The generation of MCC is so efficient that every single kinetochore signalling centre must eventually be extinguished by microtubule attachment to allow the cell to exit mitosis (Rieder *et al*, 1995; Dick & Gerlich, 2013).

This complicated inactivation process, known as SAC silencing, requires the removal of catalysts that are needed at unattached kinetochores to generate the MCC (Etemad & Kops, 2016). Two key catalysts in this regard are MAD1, which drives the first step in MCC assembly, and MPS1, the kinase responsible for recruiting and phosphorylating MAD1 as well as other components needed for MCC assembly. Kinetochore–microtubule attachment extinguishes these activities because microtubules displace MPS1 from its binding site on NDC80 (Hiruma *et al*, 2015; Ji *et al*, 2015) and at the same time they provide a highway onto which dynein motors can travel to strip MAD1 away from kinetochores (Howell *et al*, 2001; Wojcik *et al*, 2001; Mische *et al*, 2008; Sivaram *et al*, 2009). Removal of both MPS1 and MAD1 is essential for SAC silencing because if either one is artificially tethered to kinetochores, then the SAC fails to switch off and mitotic exit is blocked (Jelluma *et al*, 2010; Maldonado & Kapoor, 2011).

One key unexplained aspect of the SAC concerns the kinetochore binding sites for MAD1. MAD1 is recruited to kinetochores via an established KNL1-BUB1 pathway and, in human cells, by an additional pathway involving the ROD/ZW10/Zwilch (RZZ) complex at the kinetochore's corona (a fibrous crescent that forms around kinetochores to aid the capture of microtubules) (Luo *et al*, 2018). How exactly MAD1 is recruited to the corona and whether this pool of MAD1 can signal to the SAC are unknown. It is crucial to resolve these issues because it is ultimately the RZZ complex that is stripped by dynein to shut down the SAC, implying that this pool of MAD1 is important for MCC generation (Howell *et al*, 2001; Wojcik *et al*, 2001; Mische *et al*, 2008; Sivaram *et al*, 2009). However, the corona is positioned some distance away from MPS1 and the proposed

1 Division of Cellular Medicine, School of Medicine, University of Dundee, Dundee, UK
2 Department of Mechanistic Cell Biology, Max Planck Institute of Molecular Physiology, Dortmund, Germany
3 Oncode Institute, Hubrecht Institute—KNAW and University Medical Centre Utrecht, Utrecht, The Netherlands
*Corresponding author. E-mail: a.saurin@dundee.ac.uk

catalytic centre for MCC generation at the KNL1/MIS12/NDC80 (KMN) network. Therefore, it remains unclear how MAD1 could signal to the SAC from the corona and it is difficult to resolve this issue without knowledge of how MAD1 binds to this region.

We show here that the key mitotic kinase complex—Cyclin B1:CDK1—acts as the physical adaptor that links MAD1 to the corona. MAD1 was recently shown to recruit Cyclin B1 to kinetochores (Alfonso-Perez *et al*, 2019), and although we do see a partial reduction in kinetochore Cyclin B1 when MAD1 interaction is inhibited, the most penetrant phenotype we observe is the complete loss of corona MAD1. This unanticipated scaffolding function of Cyclin B1 is crucial for a robust SAC response, because it allows corona-tethered MAD1 to respond to low level of kinetochore MPS1 activity. This study therefore reveals how the corona pool of MAD1 signals to the SAC and it explains why MPS1 inhibition and dynein-mediated stripping of the corona are both essential for SAC silencing.

## Results

### Cyclin B1:MAD1 interaction facilitates Cyclin B1 and MAD1 localisation to unattached kinetochores

The Cyclin B:CDK1 kinase complex is a master regulator of mitosis that is activated during G2 phase of the cell cycle to initiate mitotic entry and degraded after chromosome alignment to induce mitotic exit. Analysis of endogenously tagged Cyclin B1-EYFP localisation in RPE1 cells suggested that its localisation was specifically regulated during mitosis. In particular, Cyclin B1-positive foci appeared after nuclear envelope breakdown and disappeared as mitosis progressed (Fig 1A and Movie EV1). Immunofluorescence analysis demonstrated that this localisation pattern reflects specific binding to unattached kinetochores, which is reminiscent of the checkpoint protein MAD1 (Fig 1B and C). In particular, Cyclin B1 depends on MPS1 activity to be established at this location, but thereafter it became largely insensitive to MPS1 inhibition (Fig 1D and E), as also shown previously for MAD1 (Hewitt *et al*, 2010; Etemad *et al*,

2019). Please note that in these and all subsequent quantifications, the vertical bars in the graphs represent the 95% confidence intervals, which can be used for statistical inference by eye (see Materials and Methods for full details; Cumming, 2009). To probe for MAD1 and Cyclin B1 association in cells, we recruited LacI-MAD1 to a LacO array on chromosome 1 in U2OS cells (Janicki *et al*, 2004). This was sufficient to co-recruit Cyclin B1 in a manner that was dependent on a region between amino acids 41–92 of MAD1 (Fig 1F and G). Therefore, these data are consistent with earlier reports that Cyclin B1 localises to unattached kinetochores (Bentley *et al*, 2007; Chen *et al*, 2008) in a manner that is dependent on the N-terminus of MAD1 (Alfonso-Perez *et al*, 2019; Jackman *et al*, 2020).

To determine the function of Cyclin B1 at kinetochores, we attempted to remove it from this location by knocking down endogenous MAD1 and replacing it with a Cyclin B1-binding defective mutant. However, all of the siRNAs tested only mildly reduced MAD1 protein (results not shown). This may be due to the fact that MAD1 is a very stable protein in cells because it takes over a week to fully deplete MAD1 following genetic deletion (see Rodriguez-Bravo *et al*, 2014). Therefore, to attempt to fully remove Cyclin B1 from kinetochores, we generated a MAD1α knockout cell line that retains only a MAD1β splice variant lacking exon 4 which encodes the Cyclin B1 binding region (hereafter referred to as MAD1β cells; Appendix Fig S1; Sze *et al*, 2008). Surprisingly, Cyclin B1 was reduced but still present at unattached kinetochores in MAD1β cells (Fig 1H and I). This was not due to residual interaction with MAD1β because doxycycline-inducible knockout of both MAD1α and MAD1β (McKinley & Cheeseman, 2017) completely removed MAD1 from unattached kinetochores but did not further reduce kinetochore Cyclin B1 (Fig 1J and K; note, the data shown are from 10 days of doxycycline treatment which is the minimum time it takes to fully deplete endogenous MAD1 in this system). Therefore, in contrast to a recent report (Alfonso-Perez *et al*, 2019), these data demonstrate that MAD1 contributes to Cyclin B1 kinetochore localisation, but it is not the only binding partner for Cyclin B1 at kinetochores. At least one other receptor exists that is sufficient to

---

**Figure 1. Cyclin B1:MAD1 interaction helps both proteins to localise to unattached kinetochores.**

A Endogenous Cyclin B1-YFP localisation during mitosis live in RPE1 cells (still from Movie EV1).

B, C Immunofluorescence images (B) and quantifications (C) of relative Cyclin B1 and MAD1 levels at unattached and attached kinetochores in cells arrested in STLC. Each dot represents a kinetochore, and data are from 40 kinetochore pairs (13 cells, max 5 kinetochore pairs/cell).

D, E Quantification of relative kinetochore intensities of Cyclin B1 and MAD1 in nocodazole-arrested cells (noco) treated with the MPS1 inhibitors, AZ-3146 (5 μM) or reversine (500 nM), either before (D) or after (E) mitotic entry.

F Immunofluorescence images of LacI-MAD1 and Cyclin B1 in U2OS cells containing a LacO arrays on chromosome 1.

G Live imaging of Cyclin B1-mCherry (CycB1-mCh) in LacO-U2OS cells transfected with LacI-MAD1-FL (full length: aa 1–718) or various LacI-MAD1 truncations (amino acid numbers indicated).

H, I Immunofluorescence images (H) and quantifications (I) of Cyclin B1 and MAD1 kinetochore levels in control (MAD1-WT) or MAD1β HeLa cells (two independent clones: C13 and C24) treated with nocodazole.

J, K Immunofluorescence images (J) and quantification (K) of Cyclin B1 and MAD1 kinetochore localisation in doxycycline-inducible MAD1α and MAD1β knockouts treated with or without dox for 10 days and then arrested in nocodazole. Cells were selected that had full MAD1 knockout in the doxycycline treatment (this constituted approximately 30% of cells).

L Relative kinetochore volumes occupied by Cyclin B1 and MAD1 (relative to CENP-C) in nocodazole-arrested MAD1α and MAD1β cells (calculated from experiments shown in (H, I).

Data information: For all graphs, each dot represents a cell, except panel (C) where dots represent individual kinetochores. The horizontal lines in the graphs indicate the median, and vertical bars show the 95% confidence interval. Note, when these vertical bars do not overlap, the difference is considered statistically significant at a level of at least $P < 0.05$ (see Materials and Methods). All graphs display data that are relative to the controls, which are displayed on the left side of each graph and normalised to 1. The mean level of the normalised controls is indicated by the dotted lines. (D, E, I and L) show 30 cells from 3 experiments, and K shows 40 cells from 4 experiments. Scale bars = 5 μM.

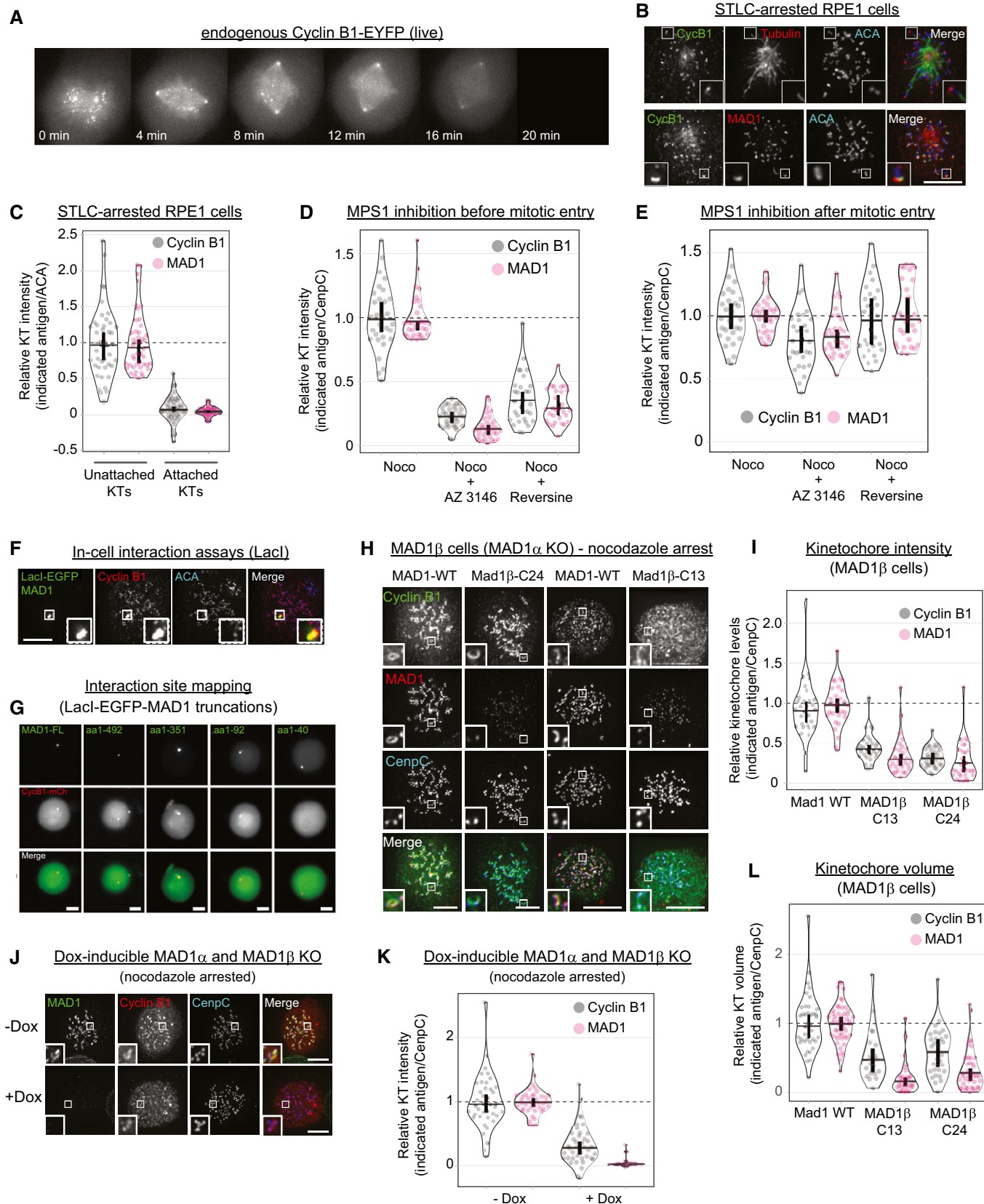

**A** endogenous Cyclin B1-EYFP (live)

0 min   4 min   8 min   12 min   16 min   20 min

**B** STLC-arrested RPE1 cells

CycB1 | Tubulin | ACA | Merge
CycB1 | MAD1 | ACA | Merge

**C** STLC-arrested RPE1 cells

Relative KT intensity (indicated antigen/ACA)

● Cyclin B1  ● MAD1

Unattached KTs    Attached KTs

**D** MPS1 inhibition before mitotic entry

Relative KT intensity (indicated antigen/CenpC)

● Cyclin B1  ● MAD1

Noco    Noco + AZ 3146    Noco + Reversine

**E** MPS1 inhibition after mitotic entry

Relative KT intensity (indicated antigen/CenpC)

● Cyclin B1  ● MAD1

Noco    Noco + AZ 3146    Noco + Reversine

**F** In-cell interaction assays (LacI)

LacI-EGFP MAD1 | Cyclin B1 | ACA | Merge

**G** Interaction site mapping (LacI-EGFP-MAD1 truncations)

MAD1-FL | aa1-492 | aa1-351 | aa1-92 | aa1-40
CycB1-mCh
Merge

**H** MAD1β cells (MAD1α KO) - nocodazole arrest

MAD1-WT | Mad1β-C24 | MAD1-WT | Mad1β-C13

Cyclin B1
MAD1
CenpC
Merge

**I** Kinetochore intensity (MAD1β cells)

Relative kinetochore levels (indicated antigen/CenpC)

● Cyclin B1  ● MAD1

Mad1 WT    MAD1β C13    MAD1β C24

**J** Dox-inducible MAD1α and MAD1β KO (nocodazole arrested)

MAD1 | Cyclin B1 | CenpC | Merge
-Dox
+Dox

**K** Dox-inducible MAD1α and MAD1β KO (nocodazole arrested)

Relative KT intensity (indicated antigen/CenpC)

● Cyclin B1  ● MAD1

- Dox    + Dox

**L** Kinetochore volume (MAD1β cells)

Relative KT volume (indicated antigen/CenpC)

● Cyclin B1  ● MAD1

Mad1 WT    MAD1β C13    MAD1β C24

**Figure 1.**

maintain substantial levels of Cyclin B1 on kinetochores in the absence of MAD1.

Although inhibiting MAD1-Cyclin B1 interaction did not abolish Cyclin B1 recruitment to kinetochores, it did cause a dramatic effect on MAD1 localisation itself. As discussed earlier, MAD1 localises to the kinetochores via two separate pathways in human cells: the KNL1-BUB1 pathway at the outer kinetochore and the RZZ pathway at the corona. Figure 1H shows that Cyclin B1 and MAD1 both bind to the corona in wild-type cells, which is present as an expanded region outside of CENP-C. However, when their interaction is prevented in MAD1β cells, it is primarily MAD1 that is lost from the corona, as evidenced by a large reduction in its kinetochore volume (Fig 1L). Therefore, this suggested that Cyclin B1 may act as a scaffold to recruit MAD1 to this region. Although MAD1 is well known to bind the corona (Buffin *et al*, 2005; Kops *et al*, 2005; Caldas *et al*, 2015; Silio *et al*, 2015; Wynne & Funabiki, 2015; Qian *et al*, 2017; Luo *et al*, 2018; Pereira *et al*, 2018; Rodriguez-Rodriguez *et al*, 2018; Sacristan *et al*, 2018; Zhang *et al*, 2019), an interaction with a corona component has never been mapped *in vitro*. In fact, the only established way to remove MAD1 from the corona is to deplete RZZ subunits, which simply abolishes corona formation altogether (Pereira *et al*, 2018; Rodriguez-Rodriguez *et al*, 2018; Sacristan *et al*, 2018). Therefore, we next sought to explore whether Cyclin B1 might be the receptor that directly recruits MAD1 to the corona.

### Cyclin B1 directly scaffolds MAD1 at the corona

We first assayed for direct MAD1 and Cyclin B1 interaction by obtaining homogeneously purified recombinant full-length MBP-MAD1:MAD2 (MBP stands for maltose-binding protein, an affinity and stabilisation tag) and Cyclin B1:CDK1 complexes and testing their interaction by size-exclusion chromatography (SEC), which separates proteins based on size and shape. When combined stoichiometrically with MBP-MAD1:MAD2, Cyclin B1:CDK1 underwent a prominent shift in elution volume and co-eluted with MAD1:MAD2, indicative of a binding interaction (Fig 2A). Early elution of MAD1:MAD2 from the SEC column reflects its high hydrodynamic radius, typical of highly elongated structures rich in coiled coil. As expected, the elution volume of MBP-MAD1:MAD2 was not affected by the interaction with Cyclin B1:CDK1.

In the absence of Cyclin B1, CDK1 did not interact directly with MAD1:MAD2 (Fig 2B). However, Cyclin B1 on its own did interact with MAD1:MAD2 (Fig 2C). Removal of residues 1–93 from MAD1 (MAD1$^{\Delta 93}$:MAD2), which are outside of the predicted coiled-coil domain of MAD1, abolished the interaction with Cyclin B1 (Fig 2D), indicating that residues 1–93 of MAD1 are necessary for the interaction. Importantly, the N-terminal region of MAD1 alone was also sufficient to bind Cyclin B1:CDK1, as revealed by SEC experiments with MBP-MAD1$^{1–92}$-SNAP and Cyclin B1:CDK1 (Fig 2E). Like the full-length MAD1:MAD2 complex, MBP-MAD1$^{1–92}$-SNAP bound to isolated Cyclin B1 but not to CDK1 (Fig EV1A and B). Therefore, MAD1 binds directly to Cyclin B1:CDK1 through a region located in the first 92 residues of MAD1.

To narrow this region down further, we performed additional truncations of MAD1$^{1–92}$, which led to the identification of a minimal Cyclin B1:CDK1 binding site in residues 41–62 of MAD1 (Fig EV1C–F). To identify determinants of the interaction between MAD1$^{41–62}$ and Cyclin B1:CDK1, we extensively mutagenised

residues in the MAD1$^{41–62}$ segment and on Cyclin B1. Charge reversals at three conserved negatively charged residues in MAD1$^{41–62}$ (E52K, E53K and E56K) abolished binding to Cyclin B1 (preprint: Allan *et al*, 2019). When introduced into the full-length MAD1:MAD2 construct, the 3EK mutation was sufficient to severely impair binding to Cyclin B1:CDK1 (Fig 2F). To identify potential binding partners on Cyclin B1 for the MAD1 residues E52, E53 and E56, we mutagenised various clusters of positively charged residues on the surface of Cyclin B1, without however identifying a sufficiently penetrant mutant (preprint: Allan *et al*, 2019). Collectively, these results indicate that MAD1 and Cyclin B1:CDK1 interact directly, and that the interaction is mediated primarily or exclusively by residues 41–62 of MAD1 and by Cyclin B1. In addition, a conserved acidic patch in this N-terminal region of MAD1 is essential for Cyclin B1 interaction (Fig 2G).

To assess the effect of inhibiting Cyclin B1:MAD1 interaction in cells, we generated doxycycline-inducible vsv-tagged MAD1-WT or MAD1-3EK HeLa cells and used these to create MAD1 knockouts via CRISPR/Cas9 (with a gRNA targeting exon 3 to knockout MAD1α and MAD1β; Fig 3A). MAD1 localisation was then assessed in nocodazole-arrested cells, which demonstrated that MAD1-WT and MAD1-3EK displayed a similar localisation pattern in early prometaphase, but only the MAD1-WT was able to localise to the corona when it formed in late prometaphase (Fig 3B). This can be seen in the kinetochore volume analysis which demonstrates that MAD1 and ZW10 kinetochore volumes increase in late prometaphase as the corona forms in MAD1-WT cells (Fig 3C and D). However, in MAD1-3EK cells, although ZW10 expands in late prometaphase, MAD1 volumes actually decrease. This represents a total drop in kinetochore MAD1-3EK levels (Fig 3B and E), which is consistent with the fact that the BUB1-dependent pool of MAD1 is reduced by PP2A as mitosis progresses (Qian *et al*, 2017). Therefore, a MAD1-3EK mutant, which is unable to bind directly to Cyclin B1, is also unable to localise to the corona in nocodazole-arrested cells. This confirms that Cyclin B1 is the scaffold that recruits MAD1 to this region of the kinetochore in human cells. When the corona pool is removed in MAD1-3EK cells, MAD1 kinetochore recruitment is reduced soon after nuclear envelope breakdown (mirroring the localisation and phosphorylation of its other kinetochore receptor, BUB1) (Nijenhuis *et al*, 2014; Qian *et al*, 2017). Note that we also generated YFP-tagged MAD1 cells to visualise its localisation live. However, YFP-MAD1-WT and YFP-MAD1-3EK were both absent from the corona, which suggests that a large N-terminal tag affects MAD1 localisation to this region (Fig EV2). This may be why removing the N-terminus of MAD1 was not reported to affect GFP/mCherry-MAD1 kinetochore localisation in previous studies (Rodriguez-Bravo *et al*, 2014; Alfonso-Perez *et al*, 2019) and why removal of the RZZ complex does not affect the kinetochore turnover of venus-MAD1 (Zhang *et al*, 2019). It is also important to note that the N-terminal vsv-tag on MAD1 is not detected at the corona by immunofluorescence (results not shown), suggesting that this region may be buried in an interaction interface.

### Corona-localised MAD1 generates a robust SAC response

The ability of Cyclin B1 to recruit MAD1 to the corona could allow Cyclin B1 to generate the signal that inhibits its own degradation. However, it is unclear whether corona-localised MAD1 can signal

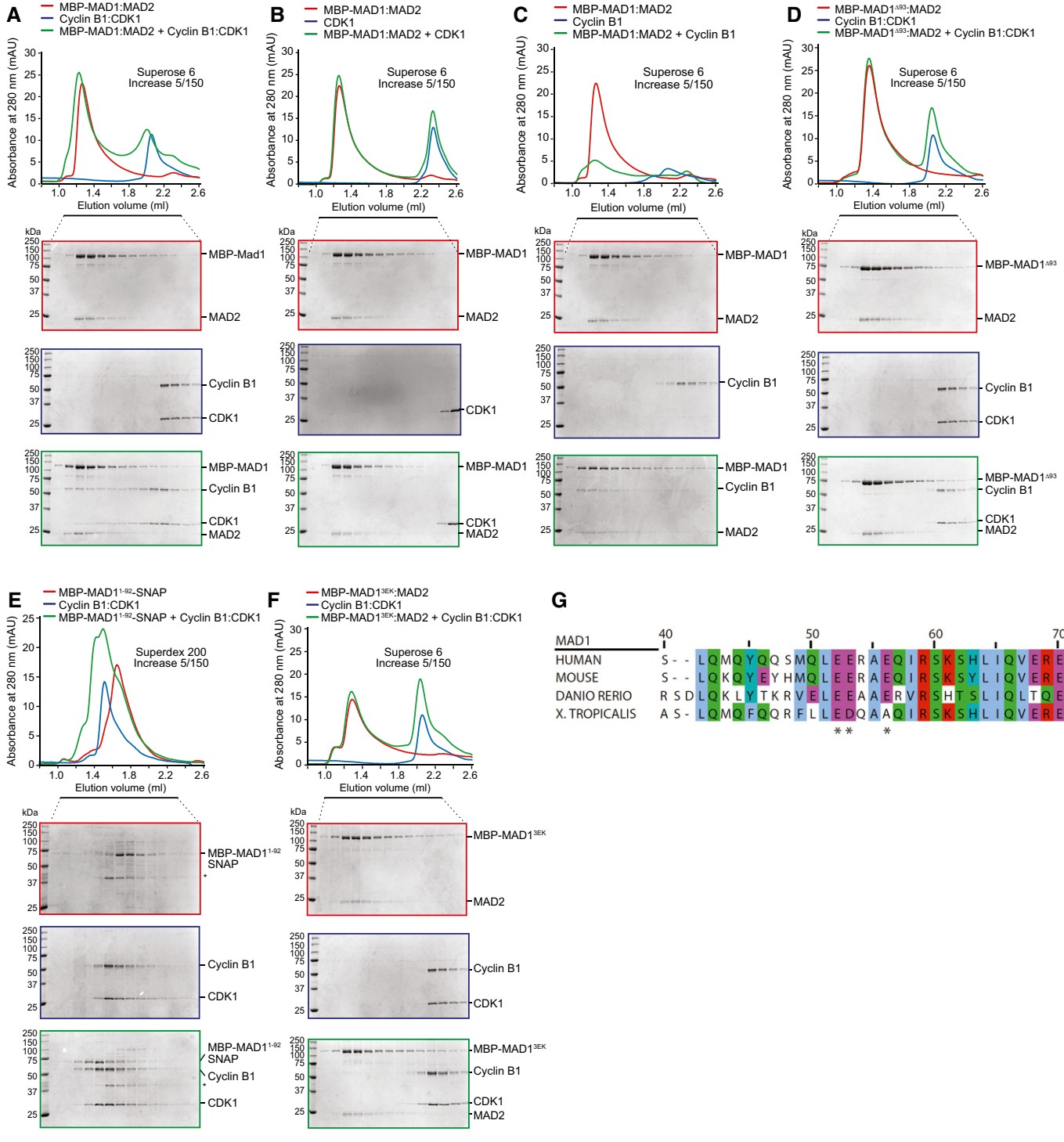

**Figure 2. Cyclin B1 and MAD1 interact directly through an N-terminal acidic patch on MAD1.**

A   Elution profiles and SDS–PAGE for SEC runs on the indicated column of the Cyclin B1:CDK1 complex (blue profile), MBP-MAD1:MAD2 (red) and their combination (green). Note that the same Cyclin B1:CDK1 elution profile and SDS–PAGE are also displayed as reference in panels (D and F) to improve clarity. For the same reason, the MBP-MAD1:MAD2 elution profile and SDS–PAGE are also displayed in panels (A–C).

B   Elution profiles and SDS–PAGE for SEC runs on the indicated column of CDK1 (blue), MBP-MAD1:MAD2 (red) and their combination (green).

C   Elution profiles and SDS–PAGE for SEC runs on the indicated column of Cyclin B1 (blue), MBP-MAD1:MAD2 (red) and their combination (green).

D   Elution profiles and SDS–PAGE for SEC runs of the Cyclin B1:CDK1 complex (blue), MBP-MAD1^Δ93:MAD2 (green) and their combination (red).

E   Elution profiles and SDS–PAGE for SEC runs of the Cyclin B1:CDK1 complex (blue), MBP-MAD1^1–92-SNAP (red) and their combination (green).

F   Elution profiles and SDS–PAGE for SEC runs of Cyclin B1:CDK1 (blue), MBP-MAD1^3EK:MAD2 (E52K, E53K and E56K mutations; red) and their combination (green). In (A-D), individual potential binding partners were combined at a concentration of 5 μM.

G   Alignment of the N-terminal region of Cyclin B1 that contains the MAD1-binding region. Numbering refers to the human MAD1 sequence. *conserved, negatively charged residues in MAD1 (E52K, E53K and E56K) required for MAD1:Cyclin B1 interaction.

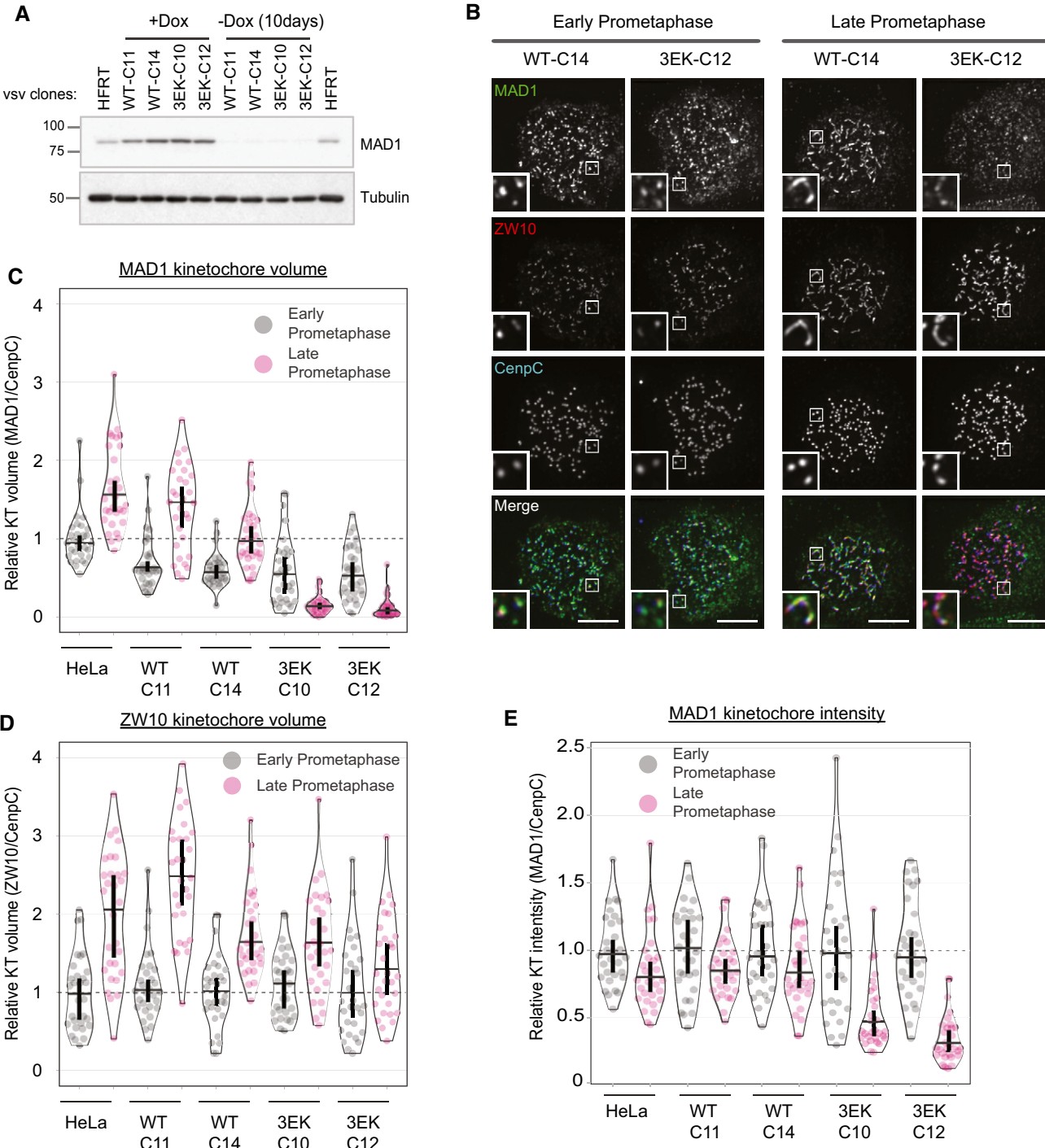

**Figure 3. Cyclin B scaffolds MAD1 at the corona.**

A Western blot analysis of indicated vsv-MAD1-WT or 3EK HeLa clones treated with or without doxycycline for 10 days.

B Immunofluorescence images showing MAD1 and ZW10 kinetochore levels in nocodazole-arrested MAD1-WT-C13 and 3EK-C14 just after nuclear envelope breakdown (early prometaphase) or later in mitosis when the chromatin is condensed (late prometaphase). Note that early and late prometaphase was defined based on the level of chromatin condensation.

C, D Relative kinetochore volumes occupied by MAD1 (C) and ZW10 (D) (relative to CENP-C) in MAD1-WT and MAD1-3EK cells in early and late prometaphase.

E Quantification of MAD1 kinetochore intensity from indicated MAD1-WT and 3EK clones treated as in (B).

Data information: For all graphs, each dot represents a cell, horizontal lines indicate the median, and vertical bars show the 95% confidence interval. Note, when these vertical bars do not overlap, the difference is considered statistically significant at a level of at least $P < 0.05$ (see Materials and Methods). All graphs display data that is relative to HeLa early prometaphase controls, which are normalised to 1. The mean level of the normalised controls is indicated by the dotted lines. 30 cells from 3 experiments. Scale bars = 5 μM.

directly to the SAC and, if it can, how this differs from the conventional KNL1-BUB1-MAD1 pathway at the outer kinetochore. One major difference is that Cyclin B1:MAD1 localisation to the corona is insensitive to MPS1 inhibition after mitotic entry (Fig 1E), whereas MPS1 activity is continually required to phosphorylate KNL1 (London *et al*, 2012; Shepperd *et al*, 2012; Yamagishi *et al*, 2012; Vleugel *et al*, 2015b) and BUB1 (London & Biggins, 2014; Mora-Santos *et al*, 2016; Faesen *et al*, 2017; Ji *et al*, 2017; Qian *et al*, 2017; Zhang *et al*, 2017) to recruit MAD1 to the outer kinetochore. To investigate this major difference between the two pathways, we tested the response of MAD1-WT and MAD1-3EK cells to MPS1 inhibition. As expected (Hewitt *et al*, 2010; Etemad *et al*, 2019), MAD1 was preserved on kinetochores following MPS1 inhibition with AZ-3146 after mitotic entry in MAD1-WT cells (Fig 4A and B). However, in stark contrast, a MAD1-3EK mutant that cannot bind the corona was completely lost from kinetochores under identical conditions (Fig 4A and B). This sensitivity to MPS1 inhibition was also mirrored by the MAD1-binding partner MAD2, a key downstream component of the MCC (Fig 4A and C). This has considerable impact on the SAC, because MAD1-3EK cells are exquisitely sensitive to MPS1 inhibition in nocodazole, as demonstrated by the fact that these cells immediately exit mitosis at a dose of AZ-3146 that can be tolerated for several hours in MAD1-WT cells (Fig 4D). These data demonstrate that Cyclin B1:MAD1 recruitment becomes insensitive to MPS1 inhibition once the corona has been established, and this subsequently allows the SAC to tolerate substantial reductions in MPS1 activity. Corona MAD1 likely also enhances SAC strength when MPS1 is not inhibited, because MAD1-3EK cells are unable to arrest as efficiently as WT cells in the presence of a CENP-E inhibitor (Fig EV3), which induces a partial checkpoint response by producing a few unattached kinetochores (Wood *et al*, 2010; Bennett *et al*, 2015).

**Cyclin B1 scaffolds MAD1 at the corona to allow the SAC to tolerate MPS1 inhibition**

As well as regulating MAD1 recruitment to the outer kinetochore, MPS1 activity is also needed to phosphorylate the C-terminal domain (CTD) of MAD1 and catalyse MCC assembly (Faesen *et al*, 2017; Ji *et al*, 2017, 2018). Therefore, we reasoned that MAD1 may still need to be phosphorylated by MPS1 to catalyse MCC assembly from the corona. Therefore, to probe this further, we raised a phospho-specific antibody to Thr716, a key MPS1-phosphorylation site on MAD1, and confirmed its specificity in cells (Fig EV4). This antibody detects a strong signal at unattached kinetochores in RPE1 and HeLa cells, and this signal is rapidly lost upon MPS1 inhibition (Figs 5A and EV4C). Importantly, in nocodazole-arrested cells, although MAD1 decorates the whole corona, the MAD1-pT716 signal is restricted primarily to the outer kinetochore around the KMN network (Figs 5A and EV4C). This suggests that either MPS1 has a limited zone of activity that is relatively confined to its anchor point on NDC80 (Hiruma *et al*, 2015; Ji *et al*, 2015) or that KMN-localised MAD1 is more resistant to dephosphorylation. How then can the corona MAD1 support the SAC following MPS1 inhibition? We hypothesised that this MAD1, which is tethered to the corona via Cyclin B1 at its N-terminus, may be able to use its predicted coiled coil to allow the CTD to reach the zone of MPS1 activity at NDC80 (Luo *et al*, 2018).

Therefore, we next set out to visualise the stable complex between MAD1:MAD2 and Cyclin B1:CDK1 by electron microscopy after low-angle metal shadowing. This highlighted the position of MAD2 near the C-terminal end of MAD1 and demonstrated that MAD1 adopted a thin elongated structure with an apparent length of ~ 66 nm (Fig 5B). Addition of the globular MBP (43 kDa) allowed the N-terminal end of full-length MAD1 to be recognised within MAD1:MAD2 complexes (Fig 5B). We then inspected a SEC fraction containing Cyclin B1:CDK1 bound to MBP-MAD1:MAD2 and identified an additional density near the N-terminal MBP in a number of complexes (Fig 5B). Thus, Cyclin B1:CDK1 binds the very end of an elongated MAD1:MAD2 complex and the opposite end, which is a substrate for MPS1 (Faesen *et al*, 2017; Ji *et al*, 2017, 2018), lies approximately 66 nm away. This observation suggests that corona-localised MAD1 can still be phosphorylated by MPS1 at the KMN network as long as the anchor point for Cyclin B1 is within ~ 66 nm of NDC80.

We hypothesised that corona-localised MAD1 could therefore help the SAC to tolerate MPS1 inhibition because, despite the fact the KNL1-MELT and BUB1 are dephosphorylated, corona-MAD1 is still preserved at kinetochores to respond to low levels of MPS1 activity. To test this hypothesis, we stained for MAD1 and MAD1-pT716 in nocodazole-arrested MAD1-WT or MAD1-3EK cells treated with a range of doses of the MPS1 inhibitor AZ-3146. Figures 5C and EV5A show that total MAD1 protein and MAD1-pT716 are removed together from kinetochores at very low doses of MPS1 inhibitor in MAD1-3EK cells. However, MAD1 is preserved at kinetochores following MPS1 inhibition in WT cells, which allows MAD1 phosphorylation to persist until much higher doses of AZ-3146. This has downstream consequence for the SAC, because the duration of mitotic arrest correlates very well with the total amount of MAD1-pT716 at kinetochores (Figs 5D and EV5B). Therefore, corona-MAD1 provides a robust SAC signal by allowing MAD1 to respond to low level of MPS1 activity and generate a prolonged mitotic arrest.

## Discussion

We show here that Cyclin B1 anchors MAD1 at the corona by binding directly to its N-terminus. This enables MAD1 to persist at this region when MPS1 activity falls, thereby allowing phosphorylation to be maintained on a key C-terminal residue in MAD1 needed for SAC catalysis (MAD1-pT716) (Faesen *et al*, 2017; Ji *et al*, 2017, 2018). We speculate that the thin elongated structure of MAD1 facilitates this process by providing the necessary reach to orient MAD2 and the MAD1 CTD towards MPS1 at the KMN network. This would explain why MAD1-pT716 is primarily restricted to the vicinity of the KMN network even though MAD1 protein is localised all over the corona (Figs 5A and EV4C).

So how exactly does corona MAD1 support the SAC and does this pool of MAD1 function differently from the canonical MAD1 pool that localises in a phospho-dependent manner to BUB1 (London & Biggins, 2014; Moyle *et al*, 2014; Mora-Santos *et al*, 2016; Qian *et al*, 2017; Zhang *et al*, 2017)? Although these two pools have previously been suggested to function independently (Silio *et al*, 2015), we favour a more integrated model whereby corona-MAD1 supports the SAC by contributing to the pool of MAD1 on the KMN

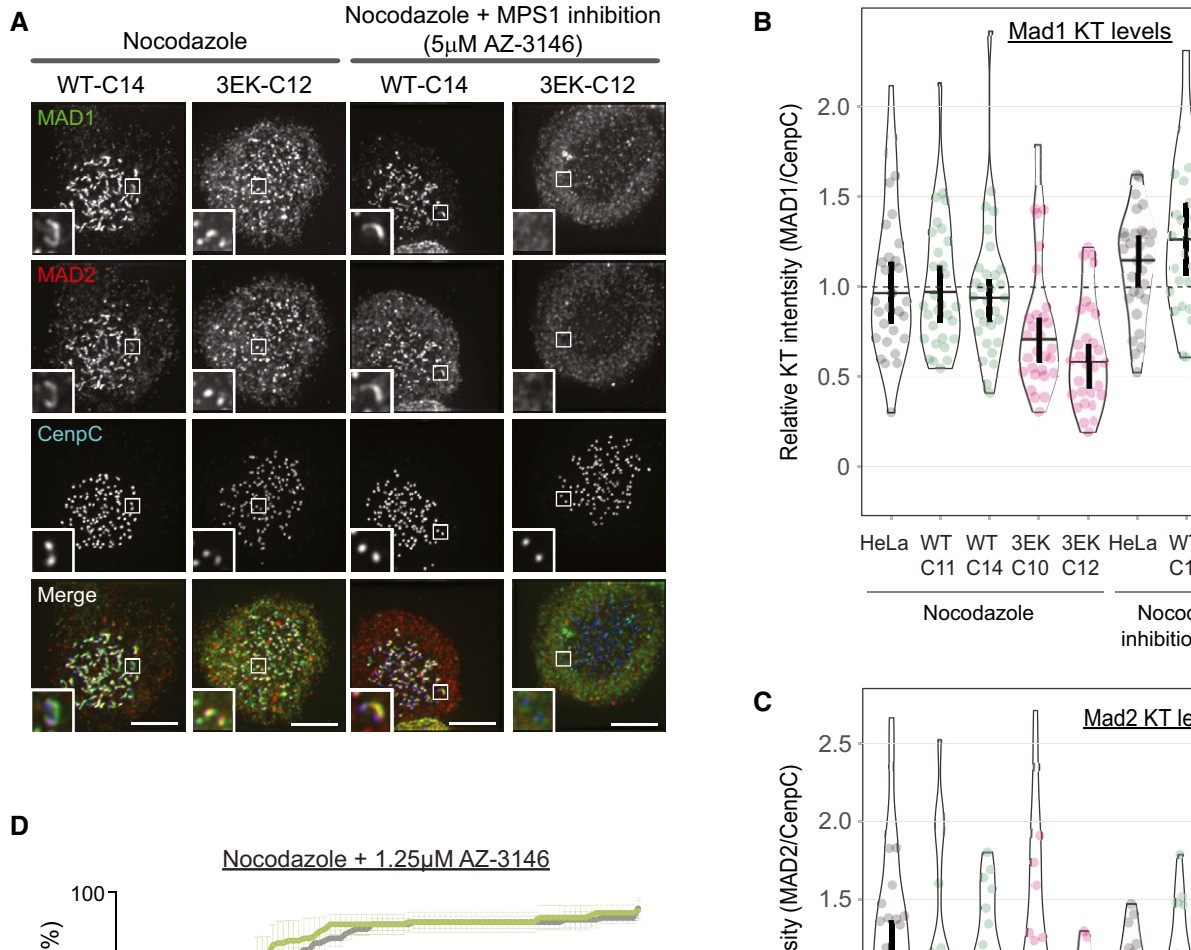

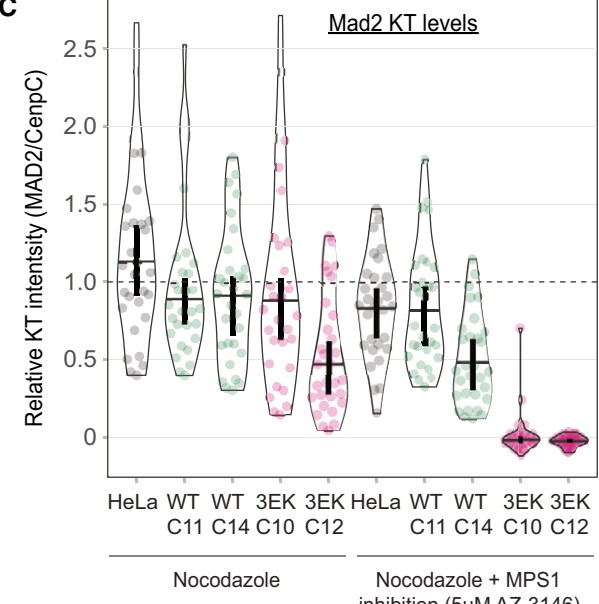

**Figure 4. Inhibiting Cyclin B1:MAD1 interaction weakens the SAC.**

A–C   Immunofluorescence images (A) and quantifications (B, C) of relative MAD1 and MAD2 kinetochore levels in indicated cells lines arrested in nocodazole and treated with/without AZ-3146 for 30 min to inhibit MPS1. In all kinetochore intensity graphs, each dot represents a cell, horizontal lines indicate the median, and vertical bars show the 95% confidence interval. Note, when these vertical bars do not overlap, the difference is considered statistically significant at a level of at least *P* < 0.05 (see Materials and Methods). All kinetochore intensity graphs display data that is relative to WT C11 controls in nocodazole, which are normalised to 1. The mean level of the normalised controls is indicated by the dotted lines. Thirty cells from 3 experiments. Scale bars = 5 μM.

D   Duration of mitotic arrest in indicated cell lines arrested in nocodazole and then treated with 1.25 μM AZ-3146. Graph shows cumulative mean (±SEM) of 3 experiments, 50 cells per condition per experiment.

network, as suggested previously by others (Zhang *et al*, 2019). We favour this hypothesis for a number of reasons: (i) corona MAD1 has been shown to facilitate MAD1-BUB1 interaction (Zhang *et al*, 2019), (ii) BUB1 is critical for cells to mount a prolonged and efficient checkpoint response (Meraldi & Sorger, 2005; Rodriguez-Rodriguez *et al*, 2018; Raaijmakers & Medema, 2019; Zhang *et al*, 2019), (iii) BUB1 is the likely catalytic centre of MCC production since it can co-localise MAD1/MAD2 together with CDC20 (Di Fiore

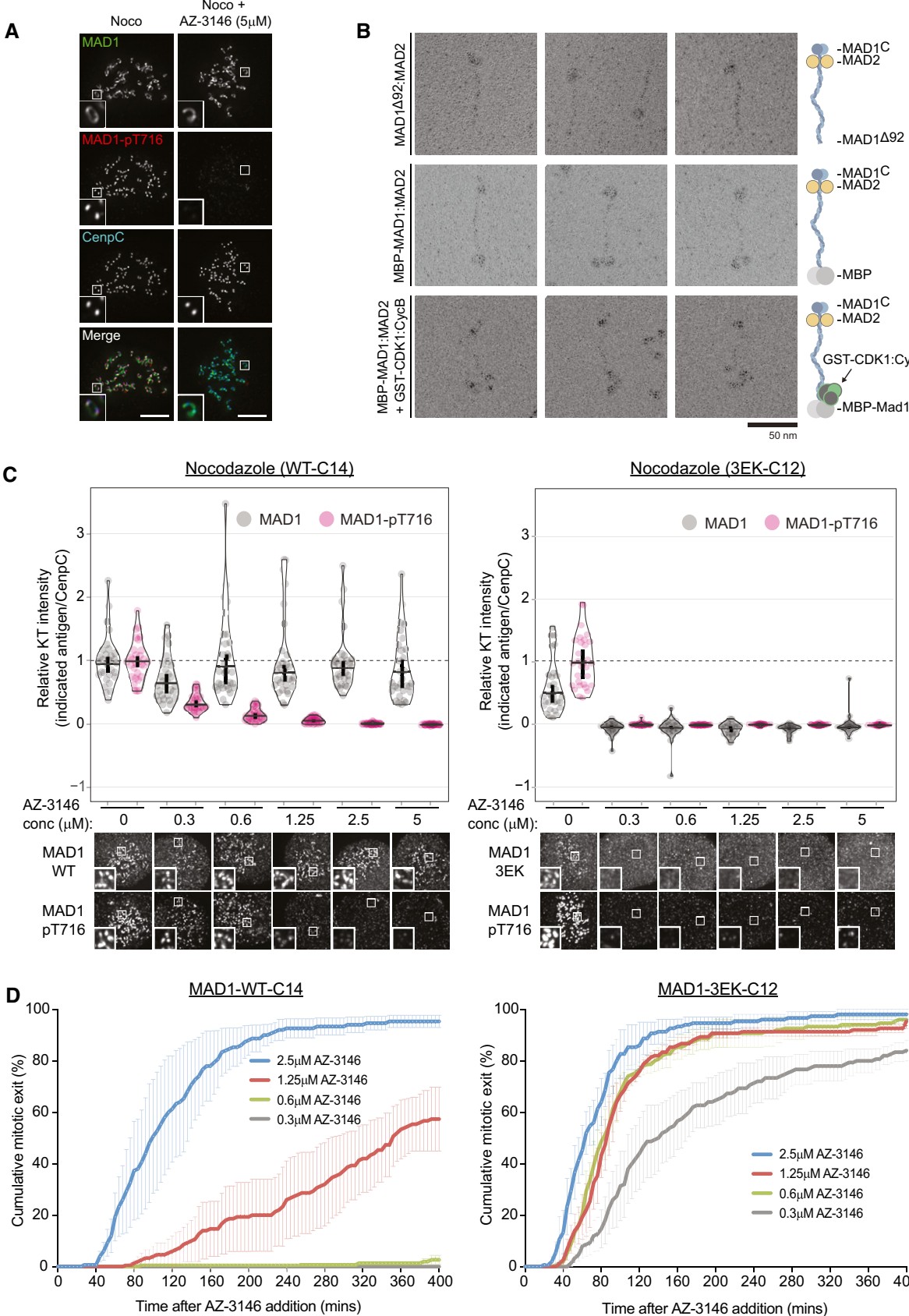

**Figure 5.**

**Figure 5.  Cyclin B1 scaffolds MAD1 at the corona to allow the SAC to tolerate MPS1 inhibition.**

A   Immunofluorescence images of MAD1 and MAD1-pT716 kinetochore levels in nocodazole-arrested RPE1 cells treated with or without AZ-3146. Scale bars = 5 μM.
B   Electron micrographs of rotary-shadowed MAD1[93–718]:MAD2 (top row), MBP-MAD1:MAD2 (middle row) and MBP-MAD1:MAD2 in complex with GST-CDK1:Cyclin B1 (bottom row). Scale bar = 50 nm.
C   Quantifications (top) and corresponding immunofluorescence images (underneath) of relative kinetochore MAD1 and MAD1-pT716 levels in nocodazole-arrested of MAD1-WT-C14 and MAD1-3EK-C12 cells treated with different doses of AZ-3146 for 30 min. MG132 was included at the time of AZ-3146 addition to prevent mitotic exit. Each dot represents a cell, horizontal lines indicate the median and error bars show the 95% confidence interval. Note, when these vertical bars do not overlap, the difference is considered statistically significant at a level of at least $P < 0.05$ (see Materials and Methods). Thirty cells from 3 experiments. Both kinetochore intensity graphs display data that is relative to the WT-C14 untreated controls, which are normalised to 1. The mean level of the normalised controls is indicated by the dotted lines.
D   Duration of mitotic arrest in MAD1-WT-C14 or MAD1-3EK-C12 cells arrested in nocodazole and then treated with indicated concentrations of AZ-3146. Note, the 1.25 μM AZ-3146 data are displayed in Fig 4D, but also included here to allow comparison with other drug doses. Graph shows cumulative mean (±SEM) of 3 experiments, 50 cells per condition per experiment.

*et al*, 2015; Vleugel *et al*, 2015a; Faesen *et al*, 2017; Ji *et al*, 2017), and (iv) the corona MAD1-pT716 that can catalyse MCC production is present in the vicinity of the KMN network where BUB1 is localised (Figs 5A and EV4C).

We can envisage two ways in which the corona and BUB1 pools of MAD1 could be inter-dependent. Firstly, their close proximity could allow MAD1 to dynamically exchange between the two locations. For example, MAD1 may release from Cyclin B1 and bind to BUB1 that is localised at KNL1. Alternatively, corona-localised MAD1, tethered via Cyclin B1 at its N-terminus, could use its elongated structure to bind directly to KNL1-localised BUB1, thereby positioning the catalytic C-terminus of MAD1 (and MAD2) towards the likely site of MCC production. In this arrangement, MAD1 would form the bridge in a tripartite Cyclin B1:MAD1:BUB1 complex that connects the corona with the KMN network. We currently favour this latter model because very recent spatial positioning data on MAD1-pT716 and MAD2 demonstrates that these signals move slightly outwards towards the RZZ complex after BUB1 depletion (preprint: Roscioli *et al*, 2019). It will be important in future to determine whether MAD1 bridges the corona and KMN network, because if it does, then this would have important implications for both SAC signalling and corona formation.

Regardless of whether corona-MAD1 contributes directly or indirectly to the KMN-localised SAC signal, its ability to be phosphorylated by low levels of MPS1 activity (Figs 5C and EV5A) explains why it must be stripped away by dynein motors following kinetochore–microtubule attachment (Howell *et al*, 2001; Wojcik *et al*, 2001; Mische *et al*, 2008; Sivaram *et al*, 2009). Otherwise, residual MPS1 activity upon bioriented kinetochores would be able to phosphorylate this pool of MAD1 and allow it to activate the SAC, possibly together with the BUB1 that remains on KNL1 (Etemad *et al*, 2019). This may explain why tethering MAD1 to kinetochores gives a prolonged mitotic arrest that is nevertheless still bypassed upon inhibition of residual MPS1 activity (Jelluma *et al*, 2010; Maldonado & Kapoor, 2011). The final model is presented in Fig 6.

There are two other recent studies that also demonstrate that MAD1-Cyclin B1 interaction is important during mitosis, but for different reasons. Alfonso-Perez *et al* (2019) demonstrated that knockdown of MAD1, or removal of its first 100 amino acids, causes a partial SAC defect in nocodazole and reduces the amount of Cyclin B1 and MPS1 on kinetochores (by approximately 75 and 50%, respectively). It should be noted, however, that the concentration of nocodazole used in these studies (0.3 μM) may be insufficient to fully depolymerise microtubules (Yang *et al*, 2009), which complicates interpretations about direct effects on the SAC and on MAD1/

Cyclin B1/MPS1 localisation to unattached kinetochores. Furthermore, removal of the N-terminal region of MAD1 has previously been shown to affect MCC assembly from the nuclear pore (Rodriguez-Bravo *et al*, 2014), which may also have contributed to the observed SAC defects. Nevertheless, the authors put forward an important hypothesis by proposing that kinetochore Cyclin B1/CDK1 may drive localised CDK1 activity to support the SAC, for example by increasing CDK1-mediated phosphorylation of MPS1 on Ser281 to enhance MPS1 localisation [see also (Hayward *et al*, 2019)]. Although kinetochore localised Cyclin B1 has not thus far been demonstrated to drive CDK1 substrate phosphorylation at kinetochores, it will be important to test this hypothesis in future. If it does regulate local CDK1 activity, then this has the potential to impact on many different processes, including the SAC. As well as MPS1 localisation, CDK1 positively and negatively regulates a number of other key enzymes at the kinetochore (for review, see Saurin, 2018), and therefore, these substrates may change dramatically upon kinetochore–microtubule attachment when Cyclin B1/MAD1 is stripped away along microtubules. Furthermore, the removal of the corona itself may depend on localised CDK1 inactivation because acute inhibition of CDK1 is known to cause premature corona detachment (Pereira *et al*, 2018; Sacristan *et al*, 2018). To address the potential importance of localised kinetochore CDK1 regulation, however, we believe it will be crucial to first identify the other receptor(s) for Cyclin B1 at the corona so that this pool can then be fully removed from unattached kinetochores.

Very recent data from Jackman *et al* (2020) has also demonstrated the importance of Cyclin B1:MAD1 interaction during mitosis, but this time in the release of MAD1 from the nuclear pore. In this study, preventing the interaction delays MAD1 accumulation at kinetochores until nuclear envelope breakdown (or just before), as well as weakening the SAC and enhancing the level of chromosomal instability (CIN). The authors use an elegant approach to mutate two key acidic residues in MAD1 (E53K/E56K) at the endogenous locus of RPE cells by CRISPR/Cas9. Although this reduces the amount of Cyclin B1 that co-precipitates with MAD1, it is possible that the additional E52K mutation included in our 3EK mutant may produce a more penetrant phenotype, since all three glutamates lie on the same face of the predicted helix in MAD1 (Jackman *et al*, 2020). Nevertheless, the subsequent results on MAD1 dissociation from the nuclear pore are entirely consistent with our data, since these focus on an earlier stage of mitosis and the authors do not examine the effects later in prometaphase at unattached kinetochores. It is likely, therefore, that Cyclin B1:MAD1 functions at the nuclear pore in early mitosis and then again at the corona following nuclear envelope

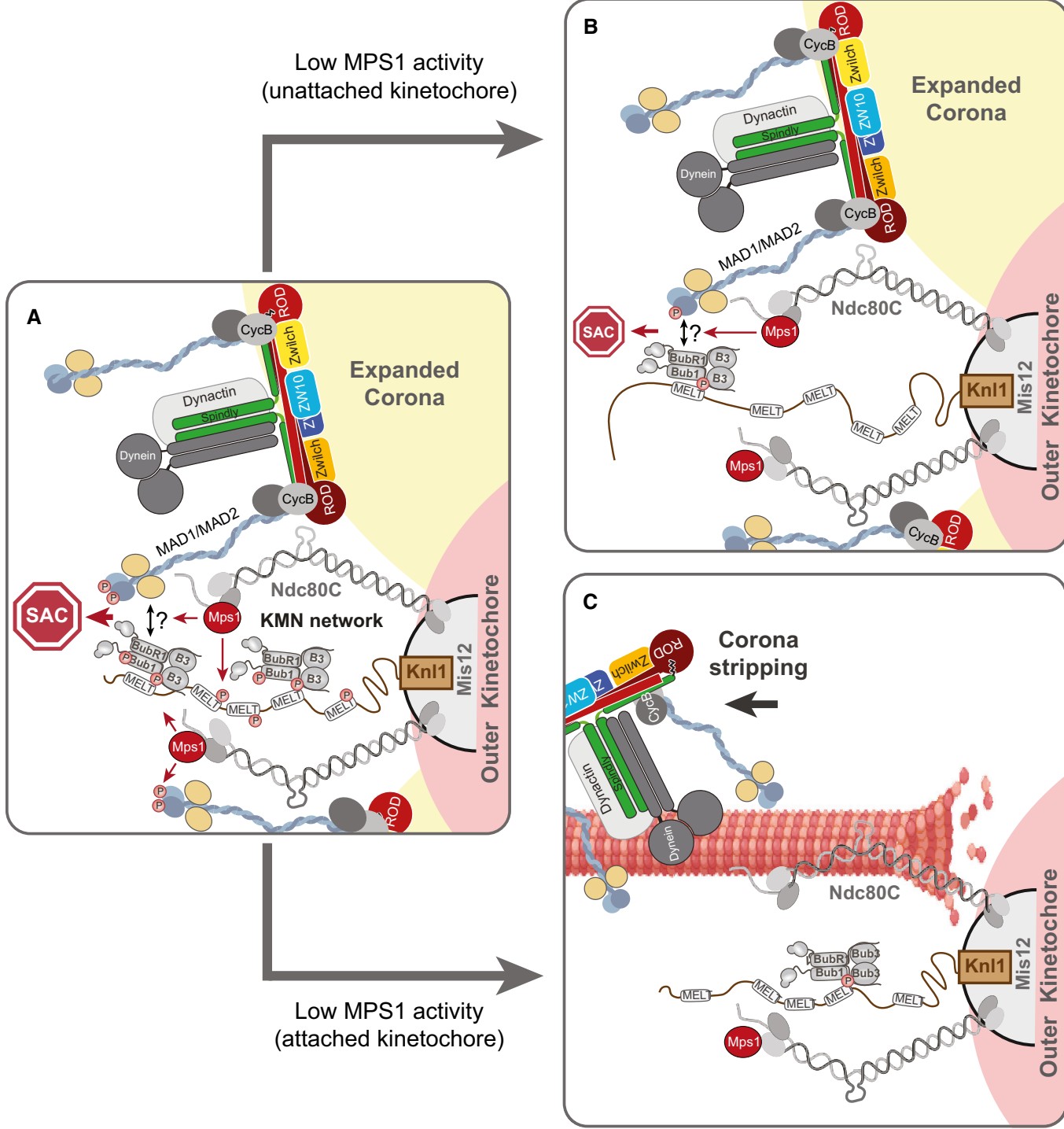

**Figure 6.  Model for how Cyclin B1:MAD1 at the corona contributes to SAC signalling.**

A   On unattached kinetochores, Cyclin B1 tethers MAD1 to the corona: note that Cyclin B (CycB) is placed next to ROD for illustrative purposes only, since the actual Cyclin B1 binding site at the corona is unknown. Corona-localised MAD1 can be phosphorylated by MPS1 on a key C-terminal residue needed for MCC catalysis (pT716), although this phosphorylation occurs mainly in the vicinity of the KMN network. MAD1 is also known to bind in a phospho-dependent manner to BUB1, but this is not illustrated because it is possible that corona-tethered MAD1 can simultaneously bind to phospho-BUB1 and drive MCC production (this is illustrated by a question mark and a double arrow).

B   If MPS1 activity is reduced without microtubule attachment, then Cyclin B1:MAD1 is maintained at the corona where it can still be phosphorylated by low-level MPS1 activity. This is sufficient to drive a SAC response, perhaps together with the residual BUB1 that remains bound to KNL1.

C   When microtubules attach, then MPS1 activity is lowered and the corona is stripped to remove all MAD1. These combined events lead to rapid silencing of the localised SAC signal.

breakdown to safeguard chromosome segregation and prevent CIN. It will be interesting to determine whether this interaction is commonly perturbed in cancer cells that frequently missegregate their chromosomes to become chromosomally unstable.

In summary, this study reveals how the main mitotic kinase, Cyclin B1/CDK1, plays a key role in scaffolding the SAC machinery to the corona. Considering that Cyclin B1 is ultimately degraded by the APC/C once the SAC has been silenced, this important scaffolding function most likely helps to ensure the SAC cannot be re-established following anaphase onset.

# Materials and Methods

### Cell culture and reagents

RPE1 cells were purchased from ATCC, and HeLa Flp-in cells were a gift from S Taylor (University of Manchester, UK) (Tighe *et al*, 2008). The RPE1 Cyclin B1-EYFP cells have been published previously (Shaltiel *et al*, 2014), as have the U2OS with LacO array on chromosome 1 (Janicki *et al*, 2004). All cells were authenticated by STR profiling (Eurofins) and screened every 4–8 weeks to ensure they were mycoplasma-free. Cells were cultured in DMEM supplemented with 9% FBS and 50 μg/ml penicillin/streptomycin, except during fluorescence time-lapse analysis, when they were cultured in Leibovitz's L-15 media (900 mg/l D+ Galactose, 5 mM Sodium Pyruvate, no phenol red). Doxycycline (1 μg/ml), STLC (S-Trityl-L-cysteine: 10 μM) and thymidine (2 mM) were purchased from Sigma-Aldrich, nocodazole (3.3 μM) from Millipore, puromycin and hygromycin B from Santa Cruz Biotechnology, MG132 (10 μM) from SelleckBio, AZ-3146 (at indicated concentrations) from Axon, rapamycin (100 nM) from LC Laboratories and reversine (at indicated concentrations) from Cayman Chemicals.

### Generation of knockout cells lines by CRISPR/Cas9 gene editing

The Cyclin B1-EYFP knockin RPE1 cell line was generated previously (Shaltiel *et al*, 2014). To generate MAD1α knockout cells (i.e. MAD1β cells), a guide RNA targeting exon 4 of MAD1 (CCGCTCCACCTGGATGAGGTGGG) was cloned into a lentiviral vector that co-expresses Cas9 and a puromycin resistance marker (pLentiCRISPRv2; Addgene #52961) to create pLentiCas9-g4-MAD1. Cyclin B1-YFP-FKBP HeLa Flp-in cells (generated by CRISPR/Cas9-mediated homologous recombination) were transfected with pLentiCas9-g4 plasmid and selected in puromycin to obtain single cell clones. These were screened for the absence of nuclear MAD1 by immunofluorescence, since only MAD1α, and not MAD1β, is localised to the nucleus in interphase (Sze *et al*, 2008), and 2 clones were subsequently validated (C13 and C24: Appendix Fig S1). Note, the original aim was to target endogenous MAD1 and replace exon 4 with an FRB cassette to lose Cyclin B1 interaction and regain it upon rapamycin addition (to induce Cyclin B1-YFP-FKBP interaction). The knockin gene editing was however unsuccessful; therefore, MAD1α knockouts were used instead. To generate doxycycline-inducible vsv- or YFP-MAD1-WT and -3EK cell lines, HeLa Flp-in cells were transfected with MAD1 in pCDNA5/FRT/TO vector (Invitrogen) together with the FLP recombinase, pOG44 (Invitrogen) using Fugene HD (Promega) according to the manufacturer's protocol.

Stable transfectants were selected in media containing 200 μg/ml hygromycin B. Subsequently, to knockout endogenous MAD1, these cells were transfected with a guide RNA targeting exon 3 of MAD1 (CTTCATCTCTCAGCGTGTGGAGG) in pLentiCRISPRv2 for 24 h and thereafter selected in puromycin for a further 24 h. Cells were then cultured continually in the presence of Dox to maintain viability after knockout of endogenous MAD1 by inducing expression of vsv- or YFP-MAD1 WT or 3EK. Individual clones were isolated and screened for loss of endogenous MAD1 by western analysis for YFP-tagged MAD1 (Fig EV2) and 10 days after removal of Dox for vsv-tagged MAD1 (Fig 3A). Two clones for each construct were validated and used subsequently (vsv-MAD1-WT: C11 & 14; vsv-MAD1-3EK: C10 & C12; YFP-MAD1-WT: C5 & C19; YFP-MAD1-3EK: C6 & C18).

### Antibodies

The following primary antibodies were used for immunofluorescence imaging (at the indicated final concentration diluted in 3% BSA in PBS): chicken α-GFP (ab13970 from Abcam, 1:5,000), guinea pig α-CENP-C (BT20278 from Caltag + Medsystems, 1:5,000), human ACA (90C-CS1058 from Fitzgerald, 1:2,000), rabbit Cyclin B1 (#12231S from Cell signalling technology, 1:1,000), mouse MAD1 (clone BB3-8, MABE867 from Millipore, 1:1,000), mouse tubulin (clone B-5-1-2 from Sigma, 1:5,000), rabbit ZW10 (ab21582 from abcam, 1:1,000) and MAD2 (A300-301A-T from Bethyl, 1:1,000). The MAD1-pT716 used in this study (MAD1-pT716-p1) was a custom rabbit polyclonal phospho-specific antibody generated by Biomatik. Secondary antibodies used for immunofluorescence were highly cross-adsorbed goat, α-chicken, α-rabbit, α-mouse or a-guinea pig coupled to Alexa Fluor 488, Alexa Fluor 568 or Alexa Fluor 647 (Thermo Fisher). The primary antibodies used for Western blotting were actin (A2066 from Sigma, 1:5,000), FKBP12 (clone H-5 from Santa Cruz, 1:250), mouse MAD1 (clone BB3-8, MABE867 from Millipore, 1:5,000), tubulin (clone 5-B-1-2 from Sigma, 1:5,000), GST (clone 8-326, MA4-004 from Thermo Fisher, 1:1,000) and Cyclin B1 (C8831 from Sigma, 1:1,000). The secondary antibodies used for Western blotting were goat α-mouse IgG HRP conjugate (170–6,516 from Bio-Rad; 1:2,000) and goat α-rabbit IgG HRP conjugate (170–6,515 from Bio-Rad; 1:5,000).

### Time-lapse analyses

For fluorescence imaging, cells were imaged in 8-well chamber slides (ibidi), on either a Zeiss Axio Observer 7 with a CMOS Orca flash 4.0 camera or a DeltaVision Elite equipped with Photometrics CascadeII:1024 EMCCD or CoolSNAP HQ (Photometrics) camera. The objectives used for fluorescent imaging were either 20×/0.8 NA or 40×/1.3NA. For brightfield imaging, cells were imaged in a 24-well plate in DMEM on a Zeiss Axiovert 200M using Hamamatsu ORCA-ER camera and controlled by Micro-manager software (open source: http://micro-manager.org), or a Zeiss Axio Observer 7 as detailed above. The air objectives used for brightfield imaging were either 10×/0.5 NA or a 20×/0.4 NA. Mitotic exit was defined by cells flattening down in the presence of nocodazole and MPS1 inhibitor.

### Immunofluorescence

This was done essentially as described previously (Smith *et al*, 2019). Cells, plated on High Precision 1.5H 12-mm coverslips

(Marienfeld), were treated and then pre-extracted with 0.1% Triton X-100 in PEM (100 mM Pipes, pH 6.8, 1 mM $MgCl_2$ and 5 mM EGTA) for 1 min before addition of 4% paraformaldehyde (PFA) in PBS for 10 min. Experiments involving MAD1-pT716 were not pre-extracted and fixed directly in 4% PFA. Coverslips were washed with PBS and blocked with 3% BSA in PBS + 0.5% Triton X-100 for at least 30 min. Thereafter, coverslips were incubated with primary antibodies overnight at 4°C, washed with PBS and incubated with secondary antibodies plus DAPI (4,6-diamidino2-phenylindole, Thermo Fisher) for an additional 2–4 h at room temperature in the dark. Coverslips were then washed with PBS and mounted on glass slides using ProLong antifade reagent (Molecular Probes). All images were acquired on a DeltaVision Core or Elite system equipped with a heated 37°C chamber, with a 100×/1.40 NA U Plan S Apochromat objective using softWoRx software (Applied precision). Images were acquired at 1 × 1 binning using a CoolSNAP HQ or HQ2 camera (Photometrics) and processed using softWorx software and ImageJ (National Institutes of Health). All immunofluorescence images displayed are maximum intensity projections of deconvolved stacks and were chosen to most closely represent the mean quantified data.

**Image quantification**

This was done essentially as described previously (Smith *et al*, 2019). For quantification of kinetochore intensities, all images to be compared directly were acquired with identical illumination settings. An ImageJ macro was used to threshold and select all kinetochores and all chromosome areas (excluding kinetochores) using the DAPI and anti-kinetochore antibody channels, as previously (Saurin *et al*, 2011). This was used to calculate the relative mean kinetochore intensity of various proteins ([kinetochores-chromosome arm intensity (test protein)]/[kinetochores-chromosome arm intensity (CENP-C)]. For the quantification of kinetochore localisation on attached vs. unattached kinetochore, cells were arrested in STLC and MAD1 intensity used to define attached (MAD1 negative) or unattached (MAD1 positive) kinetochores. The signal in the MAD1 and Cyclin B1 channel was then expressed as a percentage of CENP-C on these individual kinetochores (after normalising for surrounding background intensities). Kinetochore volumes were measured for the deconvolved stacks using the 3D object counter macro of Fiji. Threshold intensities from each channel were determined manually by opening deconvolved projections of all images to be compared (i.e. from the same experiment), equalising the channel intensities and then determining the minimum threshold that is needed to fully select all visible kinetochore staining. This threshold value was then applied to the deconvolved stacks using the 3D object counter, after cropping the image to remove any non-specific (i.e. non-kinetochore) signals (typically outside of the chromatin). The mean kinetochore volume/cell was calculated as [sum of kinetochore volume (test protein)/sum of kinetochore volume (CENP-C)]. This was then expressed as a fraction of the average value of the control sample to give the relative mean kinetochore volume per cell. Graphs were generated using either GraphPad or PlotsOfData (Postma & Goedhart, 2019).

**DNA cloning, protein expression and purification**

CDK1 and CyclinB1 cDNAs from GeneArt (Thermo Fisher) were cloned in pLIB-GST and pLIB-His8 vectors, respectively, modified

versions of the pLIB vector (Weissmann *et al*, 2016). Proteins were independently expressed in Tnao38 cells. In the case of CDK1, this was co-infected with the virus of the activator kinase CDC7. DNA sequences encoding MAD1 constructs were amplified by polymerase chain reaction (PCR) from a previously described constructs (Faesen *et al*, 2017). All the mutant versions of recombinant proteins were produced by QuikChange Mutagenesis Kit (Agilent Technologies). MAD1 full-length wild-type and 3EK mutant, as well as $MAD1^{\Delta93}$ (i.e. containing amino acids 94–718), constructs were sub-cloned in a pLIB-MBP vector, a modified version of the pLIB vector for expression in insect cells (Weissmann *et al*, 2016). The MBP-MAD1:MAD2, MBP-MAD1:MAD2$^{3EK}$ and MBP-MAD1$^{\Delta93}$:MAD2 complexes were expressed in Tnao38 cells by co-infection of the respective MAD1 baculoviruses with that of MAD2 (Faesen *et al*, 2017). MAD1 N-terminal fragments were sub-cloned into the previously generated pETDuet-MBP8His or pETDuet-MBP-SNAP-8His vectors (Liu *et al*, 2016) and were transformed into BL21(DE3) Rosetta 2 competent cells. Cells were grown in Terrific broth at 37°C to an $OD_{600}$ of about 1. Protein expression was induced by addition of 0.1 mM IPTG at 25°C, and cells were further allowed to grow overnight. All insect and bacteria cell pellets were resuspended in binding buffer (50 mM Hepes pH 7.4, 500 mM NaCl, 10% (v/v) glycerol, 2 mM TCEP), lysed by sonication and cleared by centrifugation at 10,000 × *g* for 45 min. The cleared supernatants were purified through affinity chromatography followed by gel-filtration chromatography carried out in 50 mM Hepes pH 7.4, 250 mM NaCl, 5% (v/v) glycerol and 2 mM TCEP. CDK1 was purified on glutathione sepharose (GE Healthcare) followed by GST tag removal with PreScission protease and gel-filtration chromatography on a HiLoad Superdex 200 16/60 column (GE Healthcare). Cyclin B1 was purified on HiTrap Nickel column (GE Healthcare) followed by Histidine tag removal with TEV protease and gel-filtration chromatography on a HiLoad Superdex 200 16/60 column (GE Healthcare). Cyclin B1:CDK1 complex was assembled from independently purified CDK1 and Cyclin B1 proteins mixed at 1:1.5 molar ratio and purified again on a HiLoad Superdex 200 16/60 column (GE Healthcare). MBP-MAD1 proteins were purified first on a MBP-Trap HP column (GE Healthcare) and then gel-filtered on a Superose 6 prep grade XK 16/70 or on an HiLoad Superdex 200 16/60 column (GE Healthcare). SNAP-His8 tag removal was achieved by PreScission protease cleavage and affinity chromatography on HisTrap column (GE Healthcare), before the size-exclusion chromatography step. The LacI-GFP-MAD1 fragments were generated by PCR amplification of MAD1 and insertion into the LacI-GFP vector (Vleugel *et al*, 2013). To create the guide RNA-resistant MAD1-WT and 3EK plasmids, gene blocks were synthesised that encode for amino acids 1–97 of MAD1, with or without mutations to change E52K, E53K and E56K (denoted as 3EK), and containing additional silent mutations in the gRNA sequences. These gene blocks were inserted into full-length MAD1 by Gibson assembly to replace the region encoding for amino acids 1–97.

**Analytical size-exclusion chromatography**

All samples were eluted under isocratic conditions at 4°C in SEC buffer (50 mM Hepes pH 8.0, 150 mM NaCl, 2 mM TCEP) at a flow rate of 0.12 ml/min. The elution profiles of proteins were monitored at 280 nm. To form the complex, proteins were mixed at 1:1 molar ratio and a typical concentration of 5 μM, and incubated for 30 min

on ice. The loading volume for each injection was 50 μl. SDS–PAGE, followed by Coomassie staining, was used to detect proteins.

### Low-angle metal shadowing and electron microscopy

Low-angle metal shadowing and electron microscopy were essentially performed as described (Huis In 't Veld *et al*, 2016). Fractions from size-exclusion chromatography containing protein complexes of interest were diluted 1:1 with spraying buffer (200 mM ammonium acetate and 60% glycerol) and air-sprayed onto freshly cleaved mica pieces (V1 quality, Plano GmbH). Specimens were mounted and dried in a MED020 high-vacuum metal coater (Baltec). A platinum layer of approximately 1 nm and a 7 nm carbon support layer were evaporated subsequently onto the rotating specimen at angles of 6–7° and 45°, respectively. Pt/C replicas were released from the mica on water, captured by freshly glow-discharged 400-mesh Pd/Cu grids (Plano GmbH) and visualised using a LaB$_6$ equipped JEM-1400 transmission electron microscope (JEOL) operated at 120 kV. Images were recorded at a nominal magnification of 60,000× on a 4k × 4k CCD camera F416 (TVIPS), resulting in 0.18 nm per pixel. Representative particles were manually selected using EMAN2 (Tang *et al*, 2007).

### Statistical analysis

As highlighted previously by others (Cumming, 2014; Goodman, 2016) and discussed in a statement by the American Statistical Association (Wasserstein & Lazar, 2016), there are various problems surrounding the use of *P*-values from traditional null hypothesis significance testing. We have elected instead to display 95% confidence intervals which can be used for statistical inference by eye, thus allowing the reader to easily compare any groups present on a graph (Cumming & Finch, 2005; Cumming, 2009). In this case, when the vertical bar of one condition does not overlap with the vertical bar of another condition, then the difference between the medians is statistically significant [at the level of *P* < 0.05, but often considerably smaller (Cumming, 2009)]. Therefore, the extent of overlap/separation between the vertical bars allows the reader to estimate the statistical difference between the effect sizes.

## Data availability

Source data for the quantifications displayed in graphs are provided as Expanded View Dataset EV1.

**Expanded View** for this article is available online.

### Acknowledgements

This study was funded by Cancer Research UK (C47320 to A.T.S. which also funds LA), a Leng Charitable Trust PhD studentship to M.R., and by the Max Planck Society and the European Research Council (ERC) Advanced Investigator Grant RECEPIANCE (proposal 669686) to A.M. Microscopy was carried out at the Dundee Imaging Facility. We are grateful to Iain Cheeseman for providing the inducible MAD1α and MAD1β knockout line, Stephen Taylor for the HeLa Flp-in line and Joachim Goedhart for help and advice on using PlotsOf-Data. We also thank Yahui Liu for contributing to an earlier version of this manuscript and to Nadja Harhoff for help with sub-cloning of constructs.

### Author contributions

ATS discovered the MAD1-Cyclin B1 connection whilst working in the laboratory of GJPLK (Fig 1A–C and F, G). LAA performed the majority of the cell biology experiments, with important contributions from MR. The EM of MAD1 was performed by PJH, and the *in vitro* biochemistry was performed by GC with the support of SW, under the supervision of AM. ATS wrote the manuscript with comments from all authors.

### Conflict of interest

The authors declare that they have no conflict of interest.

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
