## [Review Process File · The EMBO Journal]

Cyclin B1 scaffolds MAD1 at the kinetochore corona to activate the mitotic checkpoint

Lindsey Allan, Magda Reis, Giuseppe Ciossani, Pim Huis in 't Veld, Sabine Wohlgemuth, Geert Kops, Andrea Musacchio and Adrian Saurin.

Review timeline:

Submission date:	8 th August 2019
Editorial Decision:	13 th September 2019
Revision received:	20 th January 2020
Pre-decision correspondence	7 th February 2020
Accepted:	2 nd March 2020

Editor: Hartmut Vodermaier

Transaction Report:

1st Editorial Decision

13th September 2019

Thank you again for submitting your manuscript on Cyclin B1-MAD1 interaction for our editorial consideration. We have now received the enclosed reports of three expert referees. As you will see, all referees appreciate the technical quality of the work and generally also the overall interest of the findings, but they also raise a number of issues that would need to be adequately addressed prior to acceptance. Pending this, we would thus be happy to consider a revised manuscript further for publication in The EMBO Journal.

I should add that it is our policy to allow only a single round of major revision, making it important to diligently answer to all points raised during this round; therefore, please do not hesitate to contact me should you have any specific questions/comments regarding the reports and how to address them. Regarding the concerns of referee 2, I feel that those related to (separate) quantifications and statistical analyses would be particularly important, while further-reaching experiments such as those asked in point #4 may indeed fall beyond the scope of this revision; for point #6, I agree that earlier peer-reviewed work should already be mentioned in the introduction, while pre-published results may best be covered in the discussion section only.

REFeree REPORTS

Referee #1:

Allan and colleagues describe and characterize the direct binding of Cyclin B1 to MAD1 and investigate the role of this interaction in MAD1 localization at kinetochores and SAC signaling. MAD1 is a core component of the spindle assembly checkpoint (SAC) and its localization at kinetochores is critical for generating the SAC's soluble cell cycle inhibitor known as the mitotic

checkpoint complex (MCC). Determining the molecular determinants of MAD1 localization dynamics at kinetochores is therefore essential for understanding SAC signaling. MAD1 is known to target to two distinct sites at the kinetochore: to the KNL1/MIS12/NDC80 (KMN) network at the outer kinetochore through an interaction with BUB1 (which in turn is recruited to KNL1 via linear motifs that are phosphorylated by the checkpoint kinase MPS1), and to an expanded outermost domain (the corona), which assembles peripherally to the KMN network when kinetochores are not occupied by microtubules. The receptor(s) of MAD1 in the corona are not known, and it is unclear whether and how corona-localized MAD1 contributes to SAC signaling because the localization of MPS1, whose phosphorylation of the MAD1 C-terminus is a critical step in MCC formation, is confined to the KMN network.

The authors start by showing that Cyclin B1 and MAD1 exhibit similar localization dynamics at kinetochores and that the proteins co-localize in the corona. Using the Lac-I/Lac-O system, the authors show that MAD1(1-92) but not MAD1(1-42) is sufficient to recruit Cyclin B1 to an ectopic chromosomal site, suggesting that a MAD1-Cyclin B1 interaction may be relevant for kinetochore localization of the two proteins. To test this, the authors generate knockout cell lines in which MAD1 expression is restricted to its 'beta' splice isoform that naturally lacks the N-terminal region (encoded by exon 4) implicated in Cyclin B1 recruitment. MAD1 and Cyclin B1 levels at kinetochores are reduced (but not abolished) in MAD1 beta cells, and, importantly, MAD1 is selectively displaced from the corona but not from the outer kinetochore. The authors then embark on an impressive biochemical tour de force with purified proteins to map the region and residues in MAD1 involved in Cyclin B1 binding. Three clustered glutamate residues are identified that when mutated to lysines (3EK) abrogate the interaction with Cyclin B1 *in vitro*, and expression of the MAD1 3EK mutant in cells (in a MAD1 knockout background) displaces MAD1 from the corona but still allows MAD1 targeting to the outer kinetochore. This strongly suggests that Cyclin B1 recruits MAD1 to the corona via a direct interaction. The authors go on to show that the MAD1 3EK mutant is highly sensitive to MPS1 inhibition, consistent with the idea that the corona-pool of MAD1 allows SAC signaling to persist when MPS1 activity at the KMN network is reduced (this is relevant because the BUB1-dependent pool of MAD1 is known to be reduced by phosphatase activity as mitosis progresses). To further explore the role of corona-localized MAD1, the authors raise an antibody against a key MPS1 phosphorylation site on MAD1 (T716) and show that phospho-MAD1 is only detectable at the outer kinetochore, where MPS1 resides. EM analysis of the reconstituted MAD1-MAD2-Cyclin B1-CDK1 complex reveals that the binding site for Cyclin B1 at the MAD1 N-terminus and the MPS phosphorylation site at the MAD1 C-terminus are separated by about 66 nm. The authors propose that this highly elongated shape allows MAD1 recruited by Cyclin B1 to the corona to reach into the MPS1 active zone at the outer kinetochore to participate in SAC signaling.

This is a technically outstanding study with a compelling combination of biochemical and *in vivo* analysis. The authors identify Cyclin B1 as the MAD1 receptor at the corona, elucidate the mechanism of binding, establish a role for the corona-localized MAD1 pool in SAC signaling, and present an attractive model for how corona-localized MAD1 participates in SAC signaling at the outer kinetochore. The conclusions are supported by the data and represent a significant conceptual advance that will be of general interest to the cell cycle/mitosis community. I have a few minor comments that can be addressed without further experiments.

- 1) Panel C in Fig. 1: it would make sense to put the 'STLC-arrested RPE1 cells' label above the graph (as in Fig. 1B).
- 2) Is there a reference for the LacO array on chromosome 1 in U2OS cells that the authors use in Fig. 1F & G?
- 3) Page 1, first paragraph of results section: '...was dependent on a region between amino acids 41-92 of MAD1'. Shouldn't this be 'amino acids 43-92'?
- 4) Legend of Fig. 1C: indicate how many cells the 40 kinetochores come from and what the horizontal bars represent.
- 5) Legend of Fig. 1D, E: 'errors bars display the variation between the experimental repeats (displayed as {plus minus}SD of the experimental means)'. But then further down in the same

legend: 'horizontal lines indicate the median and vertical bars show the 95% confidence interval.'

6) Fig. S2B: can the authors comment on the interphase MAD1 staining in the MAD1 beta cells? Where is it localized?

7) Page 2, second paragraph: 'In fact, the only established way to remove MAD1 from the corona is to deplete RZZ subunits, which simply abolishes corona formation altogether.' References should be inserted after this sentence.

8) Fig. 3 & S3: the authors should comment on why the MAD1 3EK mutant was not tested in the context of the CDK1-Cyclin B1 complex, given that CDK1 seems to contribute to the affinity (compare Fig. 2A with Fig. S3B). Put another way: can one be sure that a MAD1(3EK)-MAD2 complex could not bind to the CDK1-Cyclin B complex?

9) Fig S5A, B: the authors should provide a more detailed description of how this experiment was performed. In particular, how were MAD1-WT and MAD1-S716A re-expressed in Dox(-) cells?

10) Page 7, first paragraph on the right: 'Cyclin B1/MAD1 is stripped away down microtubules.' Why 'down'?

11) Fig. S4: what is meant by light/dark? (probably shorter/longer exposure?). Please define in Fig. legend.

12) All graphs showing relative kinetochore intensities: the effects on kinetochore localization are clear. Nevertheless, some analysis of statistical significance should be included in the graphs.

Referee #2:

In this manuscript Allan and colleagues study the interaction of Cyclin B/Cdk1 and the spindle assembly checkpoint protein Mad1 at kinetochores. As previously reported by Alfonso-Perez et al in JCB at the start of the year, the authors demonstrate that Cyclin B/Cdk1 and Mad1 interact on unattached kinetochores and participate to the spindle assembly checkpoint signalling. The authors specifically show that Cyclin B and Mad1 can directly interact with each other in vitro and that Cyclin B plays an important role in the recruitment of Mad1 to the corona, a structure that arise on unattached kinetochores. Interestingly, this specific pool of Mad1 appears after an initial phase not to depend on Mps1, the main checkpoint kinase, providing a potential second, Mps1-independent SAC signalling pathway at kinetochores.

The present manuscript offers some interesting novel insights into the role and interaction of Mad1 at kinetochores, but it not entirely novel, since the key first result, i.e. that Cyclin B and the N-terminus of Mad1 are interacting at kinetochores has already been published (Alfonso-Perez et al., JCB) several months ago. Nevertheless, the study offers some new insights (the specific interaction at the corona), and presents refinements to the original model proposed by the Barr/Grüneberg laboratories in terms of SAC signalling. At the technical level, the biochemical and structural characterization of the Cyclin B/Mad1 interaction is very convincing and is a strong point of the study, and the generation of specific point mutations offers an important tool for future studies. Other aspects of the study, however, would benefit from a major revision to better support some of the main conclusions. Moreover, some of the results and conclusion should be better explained. Specifically:

Major points:

1) Robustness of the SAC response when Mad1 cannot interact with Cyclin B1.

One of the main claim of the manuscript is that the Cyclin B1-bound pool of Mad1 at the corona contributes to a robust SAC response. However, in the presence of nocodazole the cells expressing wild-type Mad1 or the 3EK Mad1 mutant remain in mitosis for the same amount of time; the only condition in which the authors see a reduction in SAC efficiency is when they treat the cells with a Mps1 inhibitor, which is a very non-physiological condition, unlikely to ever appear in a normal cell division. This conclusion could be substantially strengthened if the authors could show that the 3EK

Mad1 mutant affects the SAC response in which the checkpoint is only partially activated (e.g. with 1-3 unattached kinetochores in CENP-E depleted cells). Such experiments would also be important to resolve potential differences with the Alfonso-Perez study: as the authors themselves note, the Alfonso-Perez study reported a weaker checkpoint response at 0.3 micromolar nocodazole, which is likely to induce only few unattached kinetochores. Demonstrating that this is indeed that case, but that in conditions where all kinetochores are unattached the 3EK Mad1 mutant behaves the same would again better define the role of this Mad1 pool.

2) quantification of Mad1 and/or Cyclin B levels at kinetochores

Throughout the entire paper the authors use Mad1 and Cyc B levels on kinetochore as a key readout. At the same time the authors postulate based on their IF images the existence of two Mad1 pools: one at KT co-localizing with CENP-C and one at the corona co-localizing with the RZZ complex. To better evaluate the results, it is essential that the authors quantify the two pools separately, particularly since they postulate that different perturbations affect both pools in a distinct manner.

3) statistical analysis

Throughout the entire paper there is not a single statistical analysis of the comparative quantifications. The authors must indicate how they evaluate whether two conditions are different from each other (statistical test and p-values) and how in certain figures they compensate for the fact that they are performing multiple side-by-side comparisons (for example in Figure 4C). Without those values it is impossible to evaluate the robustness of the described results

4) "stripping of corona Mad1" is essential for SAC silencing.

In the abstract the authors claim that their study demonstrates that the stripping of the Mad1 is essential for SAC silencing once the Kn1/Bub1 branch has been silenced. However, such a model is not consistent with the recent studies (Zhang et al., EMBO J. 2019 or Rodriguez-Rodriguez, Curr Biol, 2018) showing that complete removal of Bub1 results in only a mild SAC-dependent delay in the presence of unattached kinetochores. This point should therefore be at minimum discussed; and ideally the authors could test their hypothesis by demonstrating that the combination of Bub1 removal and a 3EK Mad1 mutation leads to a complete loss of SAC signalling, even though I can see that such an experiment would take a substantial amount of time.

5) Cyclin B1 is THE Mad1 binding factor at the corona.

While the authors nicely show that Mad1 binds to the corona via Cyclin B, for the moment their quantification figures also suggest that this cannot be the sole recruitment factor, as they still observe a partial Mad1 recruitment at kinetochores when expressing Mad1-mutants that cannot interact with Cyclin B1. This again shows why a separation in a corona- and kinetochore-pool is essential for the correct interpretation of the data.

6) introduction of the current literature

For the benefit of the reader, I would recommend already presenting the Alfonso-Perez results and potentially also the Jackman et al. study results from the Pines laboratory in the introduction, as this would allow to better evaluate which aspect is novel in this present study

Minor point:

1) abstract

The sentence "We demonstrate that this allows corona-MAD1 to persist at kinetochores when MPS1 activity falls, ensuring that it can still be phosphorylated on a key C-terminal catalytic site by MPS1" in the abstract is highly cryptic, and can only be fully understood after a long and careful analysis of the results. Removing it, or simplifying it, would help to make the abstract more digestible.

Referee #3:

The spindle checkpoint protein MAD1 plays a key role in recruiting MAD2 to unattached kinetochores and further activating MAD2 for checkpoint signaling. Upon phosphorylation by the master checkpoint kinase MPS1, the BUB1-MAD1 complex scaffolds the assembly of the mitotic

checkpoint complex (MCC), which delays anaphase onset. The mechanisms by which MAD1 itself is recruited to kinetochores are complicated. BUB1 serves as an important kinetochore receptor of MAD1 and interacts with the C-terminal domain (CTD) of MAD1. The N-terminal region of MAD1 also contributes to its kinetochore localization, and this region is particularly important for localizing MAD1 to the fibrous corona. Although the localization of MAD1 to fibrous corona requires the ROD-ZW10-ZWILCH (RZZ) complex, MAD1 does not appear to interact with RZZ directly, suggesting the existence of intermediary factors.

Allan et al. have now identified one such factor as cyclin B1. They have further mapped the cyclin B1-binding motif of MAD1 to an N-terminal acidic motif. Mutating this motif abrogates the localization of MAD1 to the fibrous corona and spindle checkpoint signaling, particularly when MPS1 is partially inhibited. The authors further provide correlative data to suggest that the reliance of the MAD1 mutant on MPS1 might be due to the requirement for MAD1 CTD phosphorylation by MPS1 in checkpoint activation.

Two other groups have independently identified the MAD1-cyclin B1 interaction and characterized its roles during mitosis. Those studies had slightly different focus. The present study goes further in characterizing the biochemical and biophysical details of the interaction and provides a plausible mechanism for the heightened reliance of cells without such an interaction on MPS1 for checkpoint signaling. The development of a phospho-specific antibody that recognizes the activated form of MAD1 will enable future studies. For these reasons, the manuscript constitutes a significant enough advance for publication in a major journal.

The data are of high technical quality and generally support the authors' conclusions. The manuscript is well written and easy to follow. I only have two minor points that need to be addressed.

Minor points

- (1) The authors should measure the affinity between the N-terminal acidic motif of MAD1 and the cyclin B fragment.
- (2) In several instances in the figures, pT716 was mislabeled as pS716, and T716A as S716A. This needs to be corrected.

We have now addressed the comments from all reviewers as outlined in our point-by-point rebuttal below. One comment raised by two referees concerned statistical analysis, therefore we have first provided a general response to this below. In addition, following the initial submission of our manuscript we noticed an error in our cloning strategy to generate the gRNA-resistant MAD1-WT and MAD1-3EK cell lines. This resulted in the removal of two amino acids (aa98 and 99) during the Gibson assembly reaction that was used to generate the MAD1-WT and 3EK constructs. We therefore re-cloned both constructs, generated stable cell lines and isolated/characterised two new MAD1 knockout clones that each re-express either MAD1-WT and MAD1-3EK. We have repeated all the WT/3EK experiments with these two new clones and the conclusions are identical. We have included a side-by-side comparison of the previous and new data at the end of this rebuttal.

Finally, following the initial review, one of the authors (Dr Yahui Liu) withdrew his name and the data he had produced due to an authorship dispute raised after the paper had been submitted and the reviews had been received. The dispute was adjudicated formally by the Max-Planck-Institute of Molecular Physiology Dortmund. As a result, we have repeated the biochemical experiments on the MAD1:Cyclin B1 interaction that was contributed by Y.L. in the previous version, reaching the same conclusions as previously. Y.L. was offered authorship on the revised manuscript but he refused it. In addition, further biochemical experiments were performed during the review to substantiate our original conclusions and we now demonstrate that full length MAD1-3EK and the Cyclin B1:CDK1 complex do not interact, which was a specific issue raised by reviewer 1. The expanded set of biochemical experiments are now displayed in figures 2 and EV1.

Statistical Analysis

Reviewers 1 and 2 both commented that our data lacked statistical analysis because traditional p-values were not reported on the graphs. However, we had elected instead to display 95% confidence intervals (CI) because there are many benefits of this approach over the traditional null hypothesis significance testing (for a detailed discussion of this see (Cumming, 2014; Goodman, 2016)). Most notably, CIs can be used for statistical inference by eye, which allows the reader to easily compare any groups present on a graph (Cumming, 2009; Cumming & Finch, 2005). In this case, when the vertical bar of one condition does not overlap with the vertical bar of another condition then the difference between the medians is statistically significant (at a level of $p < 0.05$, but often considerably smaller (Cumming, 2009)). In our opinion, this ability to statistically compare the effect sizes between any conditions/treatment, is preferable to a situation where the authors display significance tests for a set of chosen conditions that are perceived to be important.

We appreciate, however, that not all readers will be familiar with this type of statistical analysis, therefore we have now added the following statement in the main text after the first quantifications:

“Please note, that in these and all subsequent quantifications, the vertical bars represent the 95% confidence intervals, which can be used for statistical inference by eye (see statistical analysis section in the methods for full details (Cumming, 2009)).”

We have then included the following expanded statement in the methods that discusses the benefits of this technique for statistical comparisons, with references to the relevant papers.

“As highlighted previously by others (Cumming, 2014; Goodman, 2016) and discussed in a statement by the American Statistical Association (Wasserstein & Lazar, 2016), there are various problems surrounding the use of p-values from traditional null hypothesis significance testing. We have elected instead to display 95% confidence intervals which can be used for statistical inference by eye, thus

allowing the reader to easily compare any groups present on a graph (Cumming, 2009; Cumming & Finch, 2005). In this case, when the vertical bar of one condition does not overlap with the vertical bar of another condition then the difference between the medians is statistically significant (at the level of $p < 0.05$, but often considerably smaller (Cumming, 2009)). Therefore, the extent of overlap/separation between the vertical bars allows the reader to estimate the statistical difference between the effect sizes”

In addition, we have now included the raw data values for all quantified data in a Source Data File

Referee #1:

Allan and colleagues describe and characterize the direct binding of Cyclin B1 to MAD1 and investigate the role of this interaction in MAD1 localization at kinetochores and SAC signaling. MAD1 is a core component of the spindle assembly checkpoint (SAC) and its localization at kinetochores is critical for generating the SAC's soluble cell cycle inhibitor known as the mitotic checkpoint complex (MCC). Determining the molecular determinants of MAD1 localization dynamics at kinetochores is therefore essential for understanding SAC signaling. MAD1 is known to target to two distinct sites at the kinetochore: to the KNL1/MIS12/NDC80 (KMN) network at the outer kinetochore through an interaction with BUB1 (which in turn is recruited to KNL1 via linear motifs that are phosphorylated by the checkpoint kinase MPS1), and to an expanded outermost domain (the corona), which assembles peripherally to the KMN network when kinetochores are not occupied by microtubules. The receptor(s) of MAD1 in the corona are not known, and it is unclear whether and how corona-localized MAD1 contributes to SAC signaling because the localization of MPS1, whose phosphorylation of the MAD1 C-terminus is a critical step in MCC formation, is confined to the KMN network.

The authors start by showing that Cyclin B1 and MAD1 exhibit similar localization dynamics at kinetochores and that the proteins co-localize in the corona. Using the Lac-I/Lac-O system, the authors show that MAD1(1-92) but not MAD1(1-42) is sufficient to recruit Cyclin B1 to an ectopic chromosomal site, suggesting that a MAD1-Cyclin B1 interaction may be relevant for kinetochore localization of the two proteins. To test this, the authors generate knockout cell lines in which MAD1 expression is restricted to its 'beta' splice isoform that naturally lacks the N-terminal region (encoded by exon 4) implicated in Cyclin B1 recruitment. MAD1 and Cyclin B1 levels at kinetochores are reduced (but not abolished) in MAD1 beta cells, and, importantly, MAD1 is selectively displaced from the corona but not from the outer kinetochore. The authors then embark on an impressive biochemical tour de force with purified proteins to map the region and residues in MAD1 involved in Cyclin B1 binding. Three clustered glutamate residues are identified that when mutated to lysines (3EK) abrogate the interaction with Cyclin B1 in vitro, and expression of the MAD1 3EK mutant in cells (in a MAD1 knockout background) displaces MAD1 from the corona but still allows MAD1 targeting to the outer kinetochore. This strongly suggests that Cyclin B1 recruits MAD1 to the corona via a direct interaction. The authors go on to show that the MAD1 3EK mutant is highly sensitive to MPS1 inhibition, consistent with the idea that the corona-pool of MAD1 allows SAC signaling to persist when MPS1 activity at the KMN network is reduced (this is relevant because the BUB1-dependent pool of MAD1 is known to be reduced by phosphatase activity as mitosis progresses). To further explore the role of corona-localized MAD1, the authors raise an antibody against a key MPS1 phosphorylation site on MAD1 (T716) and show that phospho-MAD1 is only detectable at the outer kinetochore, where MPS1 resides. EM analysis of the reconstituted MAD1-MAD2-Cyclin B1-CDK1 complex reveals that the binding site for Cyclin B1 at the MAD1 N-terminus and the MPS phosphorylation site at the MAD1 C-terminus are separated by about 66 nm. The authors propose that this highly elongated shape allows MAD1

recruited by Cyclin B1 to the corona to reach into the MPS1 active zone at the outer kinetochore to participate in SAC signaling.

This is a technically outstanding study with a compelling combination of biochemical and in vivo analysis. The authors identify Cyclin B1 as the MAD1 receptor at the corona, elucidate the mechanism of binding, establish a role for the corona-localized MAD1 pool in SAC signaling, and present an attractive model for how corona-localized MAD1 participates in SAC signaling at the outer kinetochore. The conclusions are supported by the data and represent a significant conceptual advance that will be of general interest to the cell cycle/mitosis community. I have a few minor comments that can be addressed without further experiments.

We thank reviewer 1 for the kind words and for their thorough review of our manuscript.

1) Panel C in Fig. 1: it would make sense to put the 'STLC-arrested RPE1 cells' label above the graph (as in Fig. 1B).

We agree, this has now been added.

2) Is there a reference for the LacO array on chromosome 1 in U2OS cells that the authors use in Fig. 1F & G?

Yes, the reference was originally in the methods, but we have now also placed it in the results when presenting data on the LacO cell lines.

3) Page 1, first paragraph of results section: '...was dependent on a region between amino acids 41-92 of MAD1'. Shouldn't this be 'amino acids 43-92'?

We thank the reviewer for pointing this out. The graph was actually incorrectly labelled, and the panel on the right should have read "aa1-40". This has now been corrected.

4) Legend of Fig. 1C: indicate how many cells the 40 kinetochores come from and what the horizontal bars represent.

This information has now been added.

5) Legend of Fig. 1D, E: 'errors bars display the variation between the experimental repeats (displayed as {plus minus}SD of the experimental means)'. But then further down in the same legend: 'horizontal lines indicate the median and vertical bars show the 95% confidence interval.'

This has now been corrected. All graphs now display median (horizontal bars) and 95% confidence intervals (vertical bars).

6) Fig. S2B: can the authors comment on the interphase MAD1 staining in the MAD1 beta cells? Where is it localized?

The Mad1beta is localised in the cytoplasm because Exon 4 contains a key NLS (see Sze et al, 2008). The images from Fig.S2B were taken from pre-extracted cells therefore the cytoplasmic stain is not visible. This has now been indicated in the legend for S2B (now Appendix Figure S1).

7) Page 2, second paragraph: 'In fact, the only established way to remove MAD1 from the corona is to deplete RZZ subunits, which simply abolishes corona formation altogether.' References should be inserted after this sentence.

References have now been inserted.

8) Fig. 3 & S3: the authors should comment on why the MAD1 3EK mutant was not tested in the context of the CDK1-Cyclin B1 complex, given that CDK1 seems to contribute to the affinity (compare Fig. 2A with Fig. S3B). Put another way: can one be sure that a MAD1(3EK)-MAD2 complex could not bind to the CDK1-Cyclin B complex?

This is a valid point and we thank the reviewer for raising it. We have now performed additional experiments that now demonstrate that the 3EK mutation prevents Cyclin B1:CDK1 binding to MAD1:MAD2 in the context of the full length complex (Figure 2F)

9) Fig S5A, B: the authors should provide a more detailed description of how this experiment was performed. In particular, how were MAD1-WT and MAD1-S716A re-expressed in Dox(-) cells?

This experimental approach has now been clarified in the legend of what is now Fig EV4.

10) Page 7, first paragraph on the right: 'Cyclin B1/MAD1 is stripped away down microtubules.' Why 'down'?

This has now been changed to along microtubules.

11) Fig. S4: what is meant by light/dark? (probably shorter/longer exposure?). Please define in Fig. legend.

This has now been changed to indicate short/long exposure in the figure (now Figure EV2).

12) All graphs showing relative kinetochore intensities: the effects on kinetochore localization are clear. Nevertheless, some analysis of statistical significance should be included in the graphs.

Please refer to our general point about statistics in the comments to all authors section.

Referee #2:

In this manuscript Allan and colleagues study the interaction of Cyclin B/Cdk1 and the spindle assembly checkpoint protein Mad1 at kinetochores. As previously reported by Alfonso-Perez et al in JCB at the start of the year, the authors demonstrate that Cyclin B/Cdk1 and Mad1 interact on unattached kinetochores and participate to the spindle assembly checkpoint signalling. The authors specifically show that Cyclin B and Mad1 can directly interact with each other in vitro and that Cyclin B plays an important role in the recruitment of Mad1 to the corona, a structure that arise on unattached kinetochores. Interestingly, this specific pool of Mad1 appears after an initial phase not to depend on Mps1, the main checkpoint kinase, providing a potential second, Mps1-independent SAC signalling pathway at kinetochores.

The present manuscript offers some interesting novel insights into the role and interaction of Mad1 at kinetochores, but it not entirely novel, since the key first result, i.e. that Cyclin B and the N-terminus of Mad1 are interacting at kinetochores has already been published (Alfonso-Perez et al., JCB) several months ago. Nevertheless, the study offers some new insights (the specific interaction at the corona), and presents refinements to the original model proposed by the Barr/Grüneberg laboratories in terms of SAC signalling. At the technical level, the biochemical and structural characterization of the Cyclin B/Mad1 interaction is very convincing and is a strong point of the study, and the generation of specific point mutations offers an important tool for future studies.

We would like to thank the reviewer for their thorough review of our manuscript and for their useful comments. We would also like to point out that we do not view this manuscript as simply refining the model for SAC signalling proposed by the Barr/Gruneberg laboratories. Our model certainly refines the actual MAD1:Cyclin B1 interaction, by demonstrating that it is direct and identifying the critical residues on MAD1. However, our model for how this interaction regulates the SAC is entirely different. The original Barr/Gruneberg manuscript proposes that Mad1 recruits Cyclin B to kinetochores to provide local CDK1 activity to support the SAC. Here we show that Cyclin B recruits MAD1 to the corona to scaffold the SAC signal. It may in fact be that Cyclin B1 localises MAD to the corona and elevates local CDK1 activity to support other aspects of the SAC (such as MPS1 localisation to kinetochores). However, we would like to point out that we only see a partial reduction in kinetochore Cyclin B1 following MAD1 knockout or following mutations that prevent MAD1-Cyclin B1 interaction. This differs from the Alfonso-Perez et al study, which reports a more penetrant phenotype using siRNA-mediated depletion of MAD1.

Other aspects of the study, however, would benefit from a major revision to better support some of the main conclusions. Moreover, some of the results and conclusion should be better explained. Specifically:

Major points:

1) Robustness of the SAC response when Mad1 cannot interact with Cyclin B1.

One of the main claim of the manuscript is that the Cyclin B1-bound pool of Mad1 at the corona contributes to a robust SAC response. However, in the presence of nocodazole the cells expressing wild-type Mad1 or the 3EK Mad1 mutant remain in mitosis for the same amount of time; the only condition in which the authors see a reduction in SAC efficiency is when they treat the cells with a Mps1 inhibitor, which is a very non-physiological condition, unlikely to ever appear in a normal cell division. This conclusion could be substantially strengthened if the authors could show that the 3EK Mad1 mutant affects the SAC response in which the checkpoint is only partially activated (e.g. with 1-3 unattached kinetochores in CENP-E depleted cells). Such experiments would also be important to resolve potential differences with the Alfonso-Perez study: as the authors themselves note, the Alfonso-Perez study reported a weaker checkpoint response at 0.3 micromolar nocodazole, which is likely to induce only few unattached kinetochores. Demonstrating that this is indeed that case, but that in conditions where all kinetochores are unattached the 3EK Mad1 mutant behaves the same would again better define the role of this Mad1 pool.

We thank the reviewer for pointing this out. The reason we chose to use the nocodazole + MPS1 inhibition assay is that it removes the possibility that effects on microtubule attachment impact on the SAC indirectly. Nevertheless, we have carried out the additional experiments requested, and the new data demonstrates that MAD1-3EK cells are unable to arrest as efficiently in response to CenpE inhibition (Fig. EV3).

2) quantification of Mad1 and/or Cyclin B levels at kinetochores

Throughout the entire paper the authors use Mad1 and Cyc B levels on kinetochore as a key readout. At the same time the authors postulate based on their IF images the existence of two Mad1 pools: one at KT co-localizing with CENP-C and one at the corona co-localizing with the RZZ complex. To better evaluate the results, it is essential that the authors quantify the two pools separately, particularly since they postulate that different perturbations affect both pools in a distinct manner.

This is a valid point and we have now included quantification and statistical analysis (in the form of confidence intervals) of kinetochore volumes (Figures 1L and 3C-D). This new analysis demonstrates that the ZW10 and MAD1 kinetochore pools expand during mitosis, and that MAD1 (but not ZW10) expansion is prevented when Cyclin B1 interaction is inhibited by the MAD1-3EK mutation.

3) statistical analysis

Throughout the entire paper there is not a single statistical analysis of the comparative quantifications. The authors must indicate how they evaluate whether two conditions are different from each other (statistical test and p-values) and how in certain figures they compensate for the fact that they are performing multiple side-by-side comparisons (for example in Figure 4C). Without those values it is impossible to evaluate the robustness of the described results

Please refer to our general point about statistics in the comments to all authors section.

4) "stripping of corona Mad1" is essential for SAC silencing.

In the abstract the authors claim that their study demonstrates that the stripping of the Mad1 is essential for SAC silencing once the Knl1/Bub1 branch has been silenced. However, such a model is not consistent with the recent studies (Zhang et al., EMBO J. 2019 or Rodriguez-Rodriguez, Curr Biol, 2018) showing that complete removal of Bub1 results in only a mild SAC-dependent delay in the presence of unattached kinetochores. This point should therefore be at minimum discussed; and ideally the authors could test their hypothesis by demonstrating that the combination of Bub1 removal and a 3EK Mad1 mutation leads to a complete loss of SAC signalling, even though I can see that such an experiment would take a substantial amount of time.

We thank the reviewer for this point, and we have now modified the manuscript to discuss how our data could be aligned with the above-mentioned studies from the Nilsson and Jallepalli labs. We agree that BUB1 plays a critical role in SAC signalling, although it should be pointed out that since this original review was written, it has been demonstrated that HAP1 cells completely lacking BUB1 can mount a mitotic arrest in nocodazole (Raaijmakers & Medema, 2019). Nevertheless, the question of why stripping of the corona is essential for SAC silencing is an important one that we have now discussed in the manuscript. In relation to this, it is important to note that aligned chromosomes at metaphase do not fully silence the KNL1-BUB1 pathway (see (Etemad et al., 2019)). Therefore, if corona-MAD1 is not removed, then it could still co-operate with the localised BUB1 to sustain the SAC signal from aligned kinetochores. Exactly how this corona pool of MAD1 could work together with BUB1 to support the SAC is an interesting question that will require further studies to resolve. However, we have now put forward a model that proposes that corona MAD1 may reach and bind directly to the BUB complex at kinetochores. This may explain why corona-MAD1 is needed to facilitate MAD1-BUB1 interaction (Zhang et al., 2019). The first section of the discussion has now been expanded and the model in figure 6 has been updated to include these points.

5) Cyclin B1 is THE Mad1 binding factor at the corona.

While the authors nicely show that Mad1 binds to the corona via Cyclin B, for the moment their quantification figures also suggest that this cannot be the sole recruitment factor, as they still observe a partial Mad1 recruitment at kinetochores when expressing Mad1-mutants that cannot interact with Cyclin B1. This again shows why a separation in a corona- and kinetochore-pool is essential for the correct interpretation of the data.

We believe that Cyclin B1 is essential for MAD1 corona-localisation. We find no evidence of expanded corona-localisation of MAD1 when Cyclin B interaction is inhibited, as demonstrated in the images and

in the new kinetochore volume quantifications. Therefore, Cyclin B1 is either the sole binding partner for MAD1 at the corona, or if other binding partner(s) exist, then these are insufficient to localise MAD1 to the corona without Cyclin B1.

6) introduction of the current literature

For the benefit of the reader, I would recommend already presenting the Alfonso-Perez results and potentially also the Jackman et al. study results from the Pines laboratory in the introduction, as this would allow to better evaluate which aspect is novel in this present study

We have now presented and referenced the Alfonso-Perez paper in the introduction and highlighted the novel aspects of our study.

Minor point:

1) abstract

The sentence "We demonstrate that this allows corona-MAD1 to persist at kinetochores when MPS1 activity falls, ensuring that it can still be phosphorylated on a key C-terminal catalytic site by MPS1" in the abstract is highly cryptic, and can only be fully understood after a long and careful analysis of the results. Removing it, or simplifying it, would help to make the abstract more digestible.

We thank the reviewer for pointing this out. This statement has now been simplified for clarity.

Referee #3:

The spindle checkpoint protein MAD1 plays a key role in recruiting MAD2 to unattached kinetochores and further activating MAD2 for checkpoint signaling. Upon phosphorylation by the master checkpoint kinase MPS1, the BUB1-MAD1 complex scaffolds the assembly of the mitotic checkpoint complex (MCC), which delays anaphase onset. The mechanisms by which MAD1 itself is recruited to kinetochores are complicated. BUB1 serves as an important kinetochore receptor of MAD1 and interacts with the C-terminal domain (CTD) of MAD1. The N-terminal region of MAD1 also contributes to its kinetochore localization, and this region is particularly important for localizing MAD1 to the fibrous corona. Although the localization of MAD1 to fibrous corona requires the ROD-ZW10-ZWILCH (RZZ) complex, MAD1 does not appear to interact with RZZ directly, suggesting the existence of intermediary factors.

Allan et al. have now identified one such factor as cyclin B1. They have further mapped the cyclin B1-binding motif of MAD1 to an N-terminal acidic motif. Mutating this motif abrogates the localization of MAD1 to the fibrous corona and spindle checkpoint signaling, particularly when MPS1 is partially inhibited. The authors further provide correlative data to suggest that the reliance of the MAD1 mutant on MPS1 might be due to the requirement for MAD1 CTD phosphorylation by MPS1 in checkpoint activation.

Two other groups have independently identified the MAD1-cyclin B1 interaction and characterized its roles during mitosis. Those studies had slightly different focus. The present study goes further in characterizing the biochemical and biophysical details of the interaction and provides a plausible mechanism for the heightened reliance of cells without such an interaction on MPS1 for checkpoint signaling. The development of a phospho-specific antibody that recognizes the activated form of MAD1 will enable future studies. For these reasons, the manuscript constitutes a significant enough advance for publication in a major journal.

The data are of high technical quality and generally support the authors' conclusions. The manuscript is well written and easy to follow. I only have two minor points that need to be addressed.

We thank the reviewer for the kind words and for reviewing our manuscript.

Minor points

(1) The authors should measure the affinity between the N-terminal acidic motif of MAD1 and the cyclin B fragment.

We thank the reviewer for raising this point. In response to the reviewer's concern, we tried to determine the affinity between Cyclin-B and fluorescently labelled peptides spanning wt-Mad1[41-62] or, as a negative control, 3EK-Mad1[41-62] using fluorescence anisotropy. With the peptides at 50 nM, we titrated full-length recombinant Cyclin-B or CDK1:Cyclin-B complex (and CDK1 alone as a control) from low nM to 65 μ M concentrations in a total of 15 reactions per combination. Despite the high final concentrations of Cyclin-B and the large molar excess, we did not see a meaningful increase in the polarization of the Mad1 fluorophores. This may reflect that the fluorophore retains very significant rotational freedom on the bound peptide. Alternatively, because our SEC binding assays were performed with various MBP-MAD1 fusions, MBP may facilitate the stability, solubility, or accessibility of the binding site. As explained in the rebuttal to point 8 of reviewer 1, we have now demonstrated that the 3K mutation affects CDK1:Cyclin B binding in the context of the full-length MBP-MAD1:MAD2 complex, which provides a quite definitive new control for the specificity of binding. Thus, while regretfully we cannot provide a measure of binding affinity, our biochemical analysis provides very strong evidence of the specificity of the interaction.

(2) In several instances in the figures, pT716 was mislabeled as pS716, and T716A as S716A. This needs to be corrected.

The figures have now been labelled correctly.

References

- Cumming G (2009) Inference by eye: reading the overlap of independent confidence intervals. *Stat Med* 28: 205-20
- Cumming G (2014) The new statistics: why and how. *Psychol Sci* 25: 7-29
- Cumming G, Finch S (2005) Inference by eye: confidence intervals and how to read pictures of data. *Am Psychol* 60: 170-80
- Etemad B, Vertesy A, Kuijt TEF, Sacristan C, van Oudenaarden A, Kops G (2019) Spindle checkpoint silencing at kinetochores with submaximal microtubule occupancy. *J Cell Sci* 132
- Goodman SN (2016) STATISTICS. Aligning statistical and scientific reasoning. *Science* 352: 1180-1
- Raaijmakers JA, Medema RH (2019) Killing a zombie: a full deletion of the BUB1 gene in HAP1 cells. *EMBO J* 38: e102423
- Wasserstein RL, Lazar NA (2016) The ASA Statement on p-Values: Context, Process, and Purpose. *The American Statistician* 70: 129-133
- Zhang G, Kruse T, Guasch Boldu C, Garvanska DH, Coscia F, Mann M, Barisic M, Nilsson J (2019) Efficient mitotic checkpoint signaling depends on integrated activities of Bub1 and the RZZ complex. *EMBO J* 38

Figures for Referees not shown.

Thank you for submitting your revised manuscript for our consideration. Two of the original referees have now assessed it once more, and found the new version and your responses to the original reports generally satisfactory, with referee 1 still listing a number of minor (editorial) issues that will need correction. After that, we shall be happy to publish the study in The EMBO Journal.

REFEREE REPORTS.

Referee #1 (Report for Author)

I find the revised version of the manuscript improved, and my main concerns have been addressed. The authors heroically repeated all biochemical experiments and obtained the same results, and new data is included demonstrating that MAD1(3EK)-MAD2 cannot bind Cyclin B-CDK1. The repetition of cell-based assays with new MAD1 WT and 3EK clones confirms the previous findings, and new CENP-E inhibitor experiments provide additional evidence in support of the idea that corona-localized MAD1 strengthens spindle assembly checkpoint signaling.

Regarding statistical analysis, I appreciate the authors' point about the CI overlapping method, which I assume most readers will be familiar with (i.e. I am not sure it is necessary to make the "statistical inference by eye" argument in the main text on page 2). Nonetheless, I do not quite understand the reluctance to not also perform statistical tests to compare sample medians, especially since the visual method presumably has its limits when making multiple side-by-side comparisons. Ultimately, however, the validity of the major conclusions is not in question, so I defer to journal policy regarding statistical practices.

Minor remaining issues:

- 1) Page 3: "To identify potential binding partners on Cyclin B1 for the MAD1 residues E52, E53, and E56, we mutagenized various clusters of positively charged residues on the surface of Cyclin B1, without however identifying a sufficiently penetrant mutant (unpublished data)". In the first manuscript version, the Cyclin B1 mutants were listed in Table S1, but now this is "unpublished data". I suggest that the authors either disclose what these Cyclin B mutants are in the supplement or remove this sentence from the manuscript.
- 2) Figure 1D, E: In the revised version of the figure, panel D is blown up relative to panel E. The y-axis should be the same absolute height in both panels, as in the previous version, to facilitate visual comparison of MPS1 inhibition before/after mitotic entry.
- 3) The manuscript would profit from thorough proofreading. There are still multiple typos, including the following:
 - Page 2: "statistical interference by eye" (should be "inference")
 - Legend figure 1: "D,E. 30 cells from 3 experiments". This part is redundant with the last sentence of the legend: "D, E, I and L show 30 cells from 3 experiments".
 - Figure 4B in graph: "5um" should have the correct symbol for "micro" (as in Figure 4C).
 - Page 3: "Note, that..."
 - Page 5: "its ability to be phosphorylated by low levels of MPS1 activity (Figures 5C and EV5A) explain..."
 - Page 6: "it will be important to test this hypothesis in future because..."

Referee #2 (Report for Author)

The reviewers have largely addressed my concerns, a separate quantification of the corona and kinetochore population of the described proteins could have helped, but the approach presented by the authors is convincing enough. This study would now be a very nice contribution to *EMBO Journal*.

Corresponding Author Name: Adrian T Saurin

Journal Submitted to: EMBO J

Manuscript Number: EMBOJ-2019-103180